# Neural Darwinism: A Theoretical Framework for Representation Evolution in Convolutional Neural Networks

## Abstract

We introduce a Darwinian framework that mathematically formalizes representation evolution in deep networks, viewing each neuron as an adaptive entity competing for survival during training. In this perspective, learning is governed by a unified Darwinian Score that reflects three essential dimensions of neuronal fitness—informational diversity, functional contribution, and temporal adaptability. This score induces a principled constrained optimization objective that balances model compactness with predictive fidelity, supported by new approximation guarantees showing that preserving high-fitness neurons retains the network's functional capacity. We then operationalize this framework through Neural Darwinism Culling (NDC), which serves as a practical instantiation of the Darwinian Score. NDC dynamically removes neurons with persistently low fitness while allowing high-value neurons to specialize. NDC captures the intrinsic evolutionary dynamics of neural representations: neurons with collapsed activations, negligible causal impact on loss reduction, or stagnant parameter trajectories are pruned, whereas differentiated and adaptable neurons are retained. This yields pruning decisions that are interpretable, layer-aware, and aligned with the competitive pressures naturally emerging across network depth. Experiments across diverse methodological settings demonstrate that NDC, as a direct application of the Darwinian Score, achieves substantially higher sparsity with improved generalization compared to SOTA methods, particularly under extreme compression. Ablations further confirm that the Darwinian Score is the key driver of these gains. Overall, our work provides both a general evolutionary lens for understanding representation dynamics and a practical, theory-grounded path toward efficient and adaptive deep learning.

## 1 Introduction

The rapid growth of deep learning has unlocked remarkable representational capabilities, yet at the cost of escalating computational and environmental demands Yang & Armour (2025); Chen (2025). As modern models scale, improving efficiency without sacrificing predictive fidelity has become increasingly crucial Wang (2025). Among compression techniques, pruning remains one of the most practical approaches, offering substantial reductions in memory and computation while maintaining accuracy Cho (2023); Lin (2024); Nowak (2023). However, the vast majority of pruning strategies rely on static structural heuristics—such as weight magnitudes, single-step saliency measures, or pre-defined sparsity patterns—that overlook a fundamental property of neural networks: neurons do not acquire their representational roles instantaneously. Instead, they evolve during training, differentiating in activation diversity, contributing unevenly to optimization, and adapting their parameters in response to the changing loss landscape. Ignoring these dynamics suppresses the very evolutionary processes through which meaningful representations emerge.

To address this gap, we introduce a **Darwinian mathematical framework** that conceptualizes deep learning as an evolutionary process governed by variation, selection, and adaptation. In this view, neurons act as competing entities within a dynamic ecosystem whose survival depends on their functional competence over the course of training. Central to this framework is the **Darwinian Score (DS)**, a unified fitness measure that captures the informational diversity of a neuron's activations relative to its layer population, its causal influence on loss reduction through activation–gradient

interactions, and the adaptability of its weight trajectory over time. By jointly quantifying diversity, contribution, and adaptability, the DS provides a principled assessment of a neuron's true survival value—revealing which neurons are indispensable to the function the network ultimately learns. This evolutionary formulation naturally yields a constrained optimization perspective on pruning, and we establish theoretical guarantees showing that retaining neurons with high DS preserves the network's functional approximation up to a bounded error. Building directly on this foundation, we introduce **Neural Darwinism Culling (NDC)**, which serves as a concrete operationalization—and indeed a practical application—of the DS. NDC enforces a survival-of-the-fittest mechanism during training: as the DS evolves, neurons exhibiting persistently low fitness are progressively eliminated, while highly adaptive and functionally relevant neurons are retained and encouraged to specialize. Unlike magnitude-based or static one-shot pruning, NDC makes pruning decisions that are inherently synchronized with the evolving representation dynamics of the network. In this sense, NDC is not merely another pruning heuristic but the algorithmic realization of the Darwinian viewpoint itself.

For empirical validation, we focus our study on convolutional neural networks (CNNs). CNNs remain a fundamental substrate in computer vision and offer a tractable setting for analyzing neuron-level evolution because their channel-wise structure yields clear, interpretable units of selection. Moreover, CNNs provide a stable training regime with well-understood representational hierarchies, enabling controlled comparisons and layer-wise analysis of Darwinian dynamics. While applying Darwinian principles to architectures such as Transformers or Spiking Neural Networks is a natural next step, these models introduce additional complexities—including attention-mediated interactions, token-dependent pathways, and nonstationary computational graphs—that make it substantially harder to isolate neuron-level evolutionary behavior. By contrast, CNNs offer the structural clarity and empirical stability required to validate both our theoretical framework and the operational behavior of NDC. Thus, CNNs serve not as a limitation but as an ideal testbed for establishing the core mechanisms of Darwinian representation evolution before extending the framework to more intricate architectures. Across multiple CNN architectures and datasets, NDC produces substantially higher sparsity while improving or maintaining generalization performance, remaining robust even in extreme sparsity regimes where existing methods collapse. The observed improvements directly reflect the value of modeling neuronal diversity, functional contribution, and temporal adaptability as an integrated evolutionary principle. Ablation studies further confirm that each component of the DS plays a distinct and necessary role in determining neuron survival. In summary, this work positions neural network pruning as an evolutionary selection problem, offers a rigorous mathematical framework for quantifying neuron fitness, and demonstrates that NDC is an effective and scalable application of these principles. By bridging representational dynamics and evolutionary theory, our approach provides both a conceptual understanding of how neural systems self-organize during training and a practical path toward efficient, interpretable, and adaptive deep learning.

## 2 RELATED WORK

### 2.1 TOWARDS NEURAL NETWORK COMPRESSION

Neuron compression has emerged as a central strategy for improving the efficiency of deep networks, with research spanning weight-level sparsification, structured channel or filter removal, and activation-aware criteria that leverage saliency or sensitivity to guide pruning Da Cunha & d'Amore (2023); Wright (2024); Park (2023); He & Xiao (2023); Cheng (2024). Recent works further extend this space with post-training and zero-shot approaches that compress pretrained models without costly finetuning Frantar & Alistarh (2023); Wang (2024), as well as adaptive sparsification during fine-tuning such as movement pruning Sanh (2020), and initialization-aware and lottery-ticket–style methods that identify performant subnetworks early in training Da Cunha & d'Amore (2023); Lee (2019). Beyond pruning at initialization, dynamic sparse training methods adapt network connectivity throughout training, pruning and regrowing weights to maintain sparse yet expressive subnetworks Cho (2023); Lin (2024); Nowak (2023). Other directions incorporate curvature or loss-geometry information to improve robustness after pruning Wang (2020), and joint frameworks now combine neuron-level, layerwise, and token-aware strategies for end-to-end acceleration Xiang (2023); Wang (2024); Rao (2021). Within the domain of convolutional neural networks (CNNs), which form the focus of this thesis, pruning can be broadly divided into static and dynamic regimes. Static pruning produces a fixed, globally compressed subnetwork that can be exported and deployed efficiently across inputs, with advances in structured channel pruning, hardware-aware strategies, and pruning-

at-initialization enabling practical acceleration Wright (2024); Lee (2019); Wang (2020). In contrast, dynamic pruning maintains the full model but conditionally skips filters or neurons at inference time based on input-dependent saliency or gating, thereby reducing computation on a per-example basis He & Xiao (2023); Li (2025); Liu (2018). Hybrid approaches have begun to blur the boundary between these regimes, integrating static reductions with dynamic activation control to achieve strong accuracy-efficiency trade-offs Cheng (2024). More recent studies further refine CNN pruning by explicitly modeling neuron-level relevance or activation patterns, enabling finer-grained compression while preserving generalization Park (2023); Cho (2023); Sharma (2025).

## 2.2 NEURAL DARWINIAN AND NEURAL EVOLUTION

The conceptual foundations of Neural Darwinism were laid by Edelman's seminal theory of neural group selection, which posits that competitive selection, reentrant signaling, and synaptic variation collectively drive functional specialization in the brain Edelman (1987). Recent work has extended these Darwinian principles to modern machine learning, where neuron-level selection, differentiation, and survival are increasingly formalized as mechanisms for structural efficiency and adaptive capacity. For example, theoretical and empirical studies show that pruning based on neuron sensitivity or parameter saliency can identify compact, high-performing subnetworks without full retraining Chen (2023b); Wang (2020); Lee (2019). Parallel research on evolutionary strategies and neuroevolution demonstrates that population-based mutation and selective retention can yield competitive neural architectures and initialization schemes, particularly under resource constraints or non-differentiable objectives Lange (2023); Chen (2023a); Salameh (2023); Lupu (2024). The lottery-ticket and strong-lottery investigations deepen this Darwinian perspective by formalizing how sparse survivable subnetworks are embedded within dense models and by analyzing sparsity/generalization tradeoffs across architectures Natale (2024); Frankle & Carbin (2019). Representation-focused studies reveal how activations and features evolve along trajectory-like paths during training, highlighting emergent differentiation that motivates neuron-centric interventions such as pruning, neurogenesis, or targeted regularization Kothapalli (2023); Tartaglione (2021); Chen & Ge (2024). Unifying pruning, continual learning, and evolution, several recent works adopt selective retention mechanisms inspired by Darwinian metaphors to improve lifelong adaptation, OOD robustness, and transfer efficiency Chen (2023b); Ganjdanesh (2024), while other theoretical and algorithmic advances in continual learning provide complementary foundations for stability and retention without explicitly invoking Darwinian principles Zhao (2024); Wen (2024). On the systems side, advances in evolutionary NAS, automated graph optimization, and large-scale neuroevolutionary benchmarks have lowered the engineering barrier, enabling scalable evaluation and hybrid evolutionary–gradient pipelines Lange (2023); Salameh (2023); Chen (2023a), while evolutionary strategies have also been applied in distributed and federated settings to improve system-level efficiency Rahimi (2023). Overall, these directions establish a coherent research agenda where neuron-level selection, evolutionary search, and principled sparsification converge, reviving Edelman's Neural Darwinism as a rigorous and practical framework for contemporary deep learning.

## 3 METHOD

We begin by establishing a **Darwinian Mathematical Framework** for neural representation dynamics, grounded in the observation that neurons in deep networks behave as evolutionary agents: they diversify, compete for influence, and adapt over the course of training. This framework provides the formal basis for treating each neuron as an individual entity whose survival depends on its fitness within the representational ecosystem. Within this formulation, we derive the **Darwinian Score (DS)**—a unified, theoretically motivated fitness measure that integrates three complementary axes: informational diversity, functional contribution, and evolutionary adaptability.

Building upon the Darwinian Mathematical Framework, we instantiate these principles into **Neural Darwinism Culling (NDC)**, a dynamic pruning algorithm that selectively eliminates neurons with persistently low evolutionary fitness during training. Rather than proposing yet another heuristic pruning criterion, our objective is to reveal and exploit the **intrinsic evolutionary dynamics** already present in deep learning, and to formalize how these dynamics determine which neurons are essential for maintaining the network's representational capacity.

### 3.1 PRELIMINARIES

We treat neuron activations as real-valued random variables. For neuron $i$ in layer $\ell$, let $a_i^{(\ell)}(x) \in \mathbb{R}^{H \times W}$ denote its activation for input $x \sim \mathcal{D}$. For fully-connected layers the spatial dimension reduces to a scalar. Because information-theoretic quantities appearing below are sensitive to whether one uses continuous entropy or discrete entropy, we adopt an explicit *discretization* scheme for empirical estimation. All entropy and divergence terms are computed on discretized activation histograms. We denote the histogram with $B$ bins of neuron $i$ in layer $\ell$ by $\hat{p}_{i,b}^{(\ell)}$, $b = 1, \ldots, B$, satisfying $\sum_{b=1}^{B} \hat{p}_{i,b}^{(\ell)} = 1$.

### 3.2 DARWINIAN MATHEMATICAL FRAMEWORK FOR NEURONAL SELECTION

We consider a neural system with layers $\ell = 1, \ldots, L$, where each layer contains $C_{\text{out}}^{(\ell)}$ neurons. Each neuron $i$ at layer $\ell$ is associated with activation distribution $p_i^{(\ell)}$, gradient-mediated contribution, and temporal weight trajectory $W_i^{(\ell)}(t)$. We interpret each neuron as an *individual agent* within an evolving ecosystem, and define its fitness through a unified **Darwinian Score**, which encapsulates informational diversity, functional contribution, and evolutionary adaptability within a single measure of survival potential.

Let $\{a_i^{(\ell)}(x)\}_{i=1}^{C_{\text{out}}^{(\ell)}}$ be the channel activations of layer $\ell$ evaluated on an evaluation set $\mathcal{S}$ of size $N$. Fix a common binning scheme with $B$ bins (shared across all channels in layer $\ell$), and let $n_{i,b}^{(\ell)}$ denote the integer count of activations of channel $i$ falling in bin $b$. Apply Laplace smoothing with $\varepsilon_{\text{counts}} > 0$ and define

$$\tilde{n}_{i,b}^{(\ell)} \; = \; n_{i,b}^{(\ell)} + \varepsilon_{\text{counts}}, \qquad \tilde{N}_i^{(\ell)} \; = \; \sum_{b=1}^{B} \tilde{n}_{i,b}^{(\ell)}. \tag{1}$$

The smoothed, normalized histogram for channel $i$ is

$$\tilde{p}_{i,b}^{(\ell)} \; = \; \frac{\tilde{n}_{i,b}^{(\ell)}}{\tilde{N}_i^{(\ell)}}, \qquad b = 1, \ldots, B, \tag{2}$$

and the layer-wise baseline histogram is

$$\bar{p}_b^{(\ell)} \; = \; \frac{1}{C_{\text{out}}^{(\ell)}} \sum_{j=1}^{C_{\text{out}}^{(\ell)}} \tilde{p}_{j,b}^{(\ell)}. \tag{3}$$

**Definition 3.1** (Neuron Darwinian Entropy). The Neuron Darwinian Entropy (NDE) of channel $i$ in layer $\ell$ is defined as

$$\mathsf{NDE}_i^{(\ell)} \; = \; -\sum_{b=1}^{B} \tilde{p}_{i,b}^{(\ell)} \log\left(\tilde{p}_{i,b}^{(\ell)}\right) \; + \; \mu \, D_{\text{JS}}(\tilde{p}_i^{(\ell)} \,\|\, \bar{p}^{(\ell)}),$$

where $D_{\text{JS}}$ denotes the Jensen–Shannon divergence computed on the smoothed distributions using the natural logarithm, and $\mu \geq 0$ is a scalar weight.

Intuitively, NDE quantifies the amount of *information diversity* that a neuron provides. The first term encourages neurons whose activations span a broad and balanced distribution, avoiding collapse into near-constant responses. The second term penalizes redundancy by rewarding neurons that deviate from the population baseline. Together, these two forces mirror evolutionary principles: successful individuals are not only internally diverse but also differentiated from their peers, ensuring the ecosystem (the network layer) remains heterogeneous and expressive. For clarity, we emphasize that the Neuron Darwinian Entropy (NDE) is an alignment-aware information measure, rather than a naive entropy proxy. The Shannon entropy term captures within-neuron activation diversity, while the Jensen–Shannon term measures deviation from the layer baseline, encoding population-level alignment. These jointly quantify a neuron's useful and non-redundant information contribution.

**Definition 3.2** (Activation–Gradient Contribution). For convolutional neurons (channels) we observe per-sample, per-spatial-element values $a_{i,n,h,w}^{(\ell)}$ and corresponding gradients $g_{i,n,h,w}^{(\ell)} = \partial\mathcal{L}/\partial a_{i,n,h,w}^{(\ell)}$. For a finite collection of $N$ samples we estimate

$$\widehat{\text{AGC}}_i^{(\ell)} = \left(\frac{1}{N \cdot H \cdot W} \sum_{n=1}^{N}\sum_{h=1}^{H}\sum_{w=1}^{W} \left|a_{i,n,h,w}^{(\ell)} \cdot g_{i,n,h,w}^{(\ell)}\right|^{\alpha}\right)^{1/\alpha}, \tag{4}$$

where $\alpha \geq 1$ is a robustness parameter (commonly $\alpha = 1$ or $\alpha = 2$). For fully-connected neurons the spatial sums vanish and the formula reduces to the sample mean over the $N$ examples.

The AGC quantifies *functional utility*. It captures how strongly a neuron's activation couples with the optimization signal: a neuron that frequently aligns its activation with large gradient magnitudes has a high causal impact on loss reduction. This definition deliberately integrates both forward (activation) and backward (gradient) flows, ensuring that we measure not just whether a neuron fires, but whether its firing meaningfully contributes to learning. Neurons with low AGC can be seen as weak contributors that fail to influence the network's evolutionary trajectory.

**Definition 3.3** (Neuron Adaptivity Score). Let $W_i^{(\ell)}(t)$ denote the parameter tensor of neuron $i$ at epoch $t$. We define the relative update velocity with numerical stabilization parameter $\varepsilon > 0$:

$$v_i^{(\ell)}(t) = \frac{\|W_i^{(\ell)}(t+1) - W_i^{(\ell)}(t)\|_2}{\|W_i^{(\ell)}(t)\|_2 + \varepsilon}. \tag{5}$$

Given $T$ tracked epochs (or checkpoints), the NAS is estimated as

$$\text{NAS}_i^{(\ell)} = \frac{1}{T}\sum_{t=1}^{T} v_i^{(\ell)}(t) + \sigma \cdot \text{Var}_{t=1}^{T}\left(v_i^{(\ell)}(t)\right), \tag{6}$$

with $\sigma \geq 0$ a hyperparameter controlling the weight of variability.

NAS measures the evolutionary adaptability of a neuron. The average velocity term captures whether the neuron's parameters meaningfully evolve during training, while the variance term detects flexibility to nonstationary signals. Biologically, this reflects the difference between neurons that remain static (rigid and maladaptive) versus those that actively adapt to changing environments. A neuron with high NAS can be considered more "evolutionarily fit," as it demonstrates both learning capacity and responsiveness to the broader optimization dynamics.

**Definition 3.4** (Darwinian Score). For a fixed layer $\ell$, let the raw metric values for neuron $i$ be $S_{i,s}^{(\ell)}$ for $s \in \{\text{NDE, AGC, NAS}\}$. Define the robust, layer-wise normalized score

$$\tilde{S}_{i,s}^{(\ell)} = \frac{S_{i,s}^{(\ell)} - \text{median}_j S_{j,s}^{(\ell)}}{\text{IQR}_j\left(S_{j,s}^{(\ell)}\right) + \varepsilon}, \tag{7}$$

where IQR is the interquartile range and $\varepsilon > 0$ prevents division by zero. To map normalized scores into the positive domain required for log-domain aggregation, let

$$\hat{S}_{i,s}^{(\ell)} = \max\left(\tilde{S}_{i,s}^{(\ell)}, \delta_{\text{floor}}\right) + \varepsilon_{\text{pos}}, \tag{8}$$

with $\delta_{\text{floor}} \geq 0$ and $\varepsilon_{\text{pos}} > 0$ small constants. The *Darwinian score* of neuron $i$ in layer $\ell$ is then defined by the (log-domain) weighted geometric aggregation

$$\mathcal{I}_i^{(\ell)} = \exp\left(\eta \sum_{s \in \{\text{NDE,AGC,NAS}\}} w_s \log\left(\hat{S}_{i,s}^{(\ell)}\right)\right), \tag{9}$$

where $w_s \geq 0$ are metric weights and $\eta > 0$ is a global scaling factor.

This multiplicative structure enforces a strict Darwinian criterion: a neuron survives only if it performs well across the three dimensions. Unlike additive schemes, which allow a high score in one dimension to compensate for deficiencies in another, our multiplicative design ensures that weakness in diversity, contribution, or adaptability suffices for elimination. In other words, neurons must jointly exhibit information richness, functional utility, and adaptivity in order to be deemed indispensable.

### 3.2.1 DYNAMIC DARWINIAN SCORE

To investigate how selective pressures manifest throughout the hierarchy of a deep convolutional architecture, we tracked the evolution of the Darwinian Score (DS) for every neuron during the training of VGG-16 on CIFAR-10. Figure 1 depicts the DS trajectories in representative layers, where each curve corresponds to a single neuron and is recorded at every optimization step. A consistent pattern emerges across depths: neurons undergo a rapid differentiation phase within the first 5,000 steps, followed by gradual stabilization into quasi-stationary regimes. Interestingly, earlier layers exhibit lower absolute DS values with tighter convergence bands, suggesting stronger homogenization of representational roles, while deeper layers sustain higher DS levels and broader variance, indicative of more persistent competition and specialization. This progression aligns with the principle of Neural Darwinism, where variation, competition, and selective retention become increasingly pronounced in deeper representational stages. These findings provide empirical evidence that the dynamics of neuron-level survival and specialization are not uniform across depth but rather amplify as representations move toward higher-level abstractions.

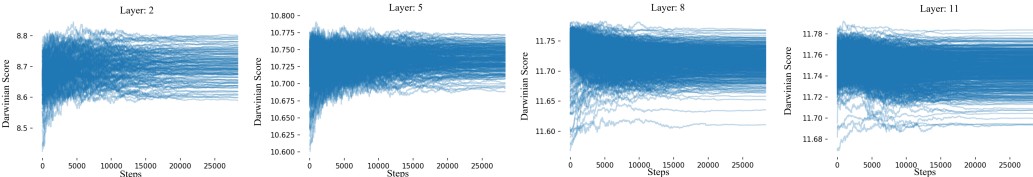

Figure 1: Dynamic of Darwinian Score at different layers in the VGG-16 network trained on the CIFAR-10 dataset. Each of the lines records the DS of a neuron during training every step.

## 3.3 PRUNING AS CONSTRAINED OPTIMIZATION

Formally, pruning solves

$$\min_{\mathcal{M}} \quad \mathcal{L}\big(f(x; \Theta \odot \mathcal{M}), y\big) \quad \text{s.t.} \quad \|\mathcal{M}\|_0 \leq \kappa, \tag{10}$$

where $\mathcal{M}$ is a binary mask over neurons and $\kappa$ the survival budget. We keep neurons with the largest $\mathcal{I}_i^{(\ell)}$.

This formulation highlights pruning as a structured selection process akin to natural selection: given limited resources (budget $\kappa$), the ecosystem retains only those individuals with maximal Darwinian fitness. By construction, the survivors are those that are simultaneously diverse, contributive, and adaptive, ensuring that the reduced subnetwork remains expressive and robust.

**Lemma 3.5** (Importance Concentration). *If the distribution of $\mathcal{I}_i^{(\ell)}$ in layer $\ell$ is $\delta$-concentrated, i.e., the top-$\rho$ neurons account for at least $(1 - \delta)$ of cumulative importance, then pruning $(1 - \rho)$ fraction of neurons increases error by at most $O(\delta)$.*

**Theorem 3.6** (Darwinian preservation). *Let $f$ be the network mapping and for each layer $\ell$ let the nonnegative importance scores $\{\mathcal{I}_i^{(\ell)}\}_{i=1}^{C_{\text{out}}^{(\ell)}}$ be normalized as*

$$\tilde{\mathcal{I}}_i^{(\ell)} = \frac{\mathcal{I}_i^{(\ell)}}{\sum_{j=1}^{C_{\text{out}}^{(\ell)}} \mathcal{I}_j^{(\ell)}}.$$

*Fix per-layer retention fractions $\rho^{(\ell)} \in (0, 1]$ and suppose that for every layer $\ell$ there exists a retained index set $S_\ell$ with $|S_\ell| = \rho^{(\ell)} C_{\text{out}}^{(\ell)}$ such that*

$$\sum_{i \in S_\ell} \tilde{\mathcal{I}}_i^{(\ell)} \geq 1 - \delta^{(\ell)}.$$

*Assume the following* expectation contribution comparability *condition: for each layer $\ell$ there exists $\kappa_\ell \geq 0$ such that for any subset $T \subseteq \{1, \ldots, C_{\text{out}}^{(\ell)}\}$, the post-activation layer outputs $z^{(\ell)}(x)$ and $z^{(\ell),(-T)}(x)$ (after zeroing neurons in $T$) satisfy*

$$\mathbb{E}_{x \sim \mathcal{D}} \Big[ \big\| z^{(\ell)}(x) - z^{(\ell),(-T)}(x) \big\|_p \Big] \leq \kappa_\ell \sum_{i \in T} \tilde{\mathcal{I}}_i^{(\ell)}. \tag{11}$$

*Suppose further that layer-$\ell$ post-activation outputs are uniformly bounded as $\|z^{(\ell)}(x)\|_\infty \leq A_\ell$ for all $x \sim \mathcal{D}$, and that the mapping from layer-$\ell$ outputs to the final network output is $L^{(\ell)}$-Lipschitz with respect to the same norm $\|\cdot\|_p$. Define*

$$C_\ell = L^{(\ell)} A_\ell \kappa_\ell.$$

*Then the pruned network $\hat{f}$ obtained by retaining only the neurons in $\bigcup_\ell S_\ell$ satisfies*

$$\mathbb{E}_{x \sim \mathcal{D}} \left[ \|f(x) - \hat{f}(x)\|_p \right] \leq \sum_{\ell=1}^L C_\ell \, \delta^{(\ell)}.$$

These results provide formal guarantees that pruning guided by Darwinian importance preserves function approximability. In essence, NDE ensures informational coverage, AGC ensures optimization relevance, and NAS ensures adaptability. Taken together, their multiplicative integration guarantees that the surviving subnetwork retains the key drivers of generalization.

### 3.4 APPLICATION: OUR NEURAL DARWINISM CULLING

#### 3.4.1 NDC IN CONVOLUTIONAL LAYERS

In convolutional networks, neurons are naturally grouped as channels. For a convolutional layer $\ell$ with $C_{\text{out}}^{(\ell)}$ output channels, each neuron corresponds to a filter $W_i^{(\ell)} \in \mathbb{R}^{C_{\text{in}}^{(\ell)} \times k \times k}$. We compute Darwinian metrics at the channel level. Specifically, $\text{NDE}_i^{(\ell)}$ is estimated from the spatially aggregated activations $a_i^{(\ell)}(x) \in \mathbb{R}^{H \times W}$, treating their empirical distribution as $p_i^{(\ell)}$. The $\text{AGC}_i^{(\ell)}$ metric is obtained from the product of channel activations and gradients aggregated over spatial dimensions. Finally, $\text{NAS}_i^{(\ell)}$ is tracked by monitoring the evolution of the filter weights $W_i^{(\ell)}$ across epochs. This design ensures that pruning respects the convolutional structure: entire filters (channels) are retained or discarded, thereby preserving tensor consistency in subsequent layers.

#### 3.4.2 NDC IN LINEAR LAYERS

For fully linear layers, each neuron $i$ corresponds to a row vector $W_i^{(\ell)} \in \mathbb{R}^{C_{\text{in}}^{(\ell)}}$. Here, Darwinian fitness is evaluated at the single-neuron granularity. The metric $\text{NDE}_i^{(\ell)}$ is computed directly from the scalar activations across samples. The $\text{AGC}_i^{(\ell)}$ score couples activations with their gradient signals to measure functional contribution. Meanwhile, $\text{NAS}_i^{(\ell)}$ captures the temporal dynamics of weight updates, reflecting adaptability of individual neurons. This formulation allows NDC to adapt seamlessly to dense architectures while maintaining consistency with the convolutional setting.

#### 3.4.3 PRUNING STRATEGY

Having computed the Darwinian quantities for each neuron, NDC first applies robust normalization and log-domain aggregation to consolidate them into a unified Darwinian score. $\mathcal{I}_i^{(\ell)}$ (Algorithm 1 in Appendix). Pruning is then carried out in a structured, layer-aware manner: neurons within each layer $\ell$ are ranked by $\mathcal{I}_i^{(\ell)}$, the top-$\kappa^{(\ell)}$ are preserved, and the remainder are suppressed through a binary mask $\mathcal{M}^{(\ell)}$, resulting in the pruned subnetwork $\hat{f}$. This design ensures both intra-layer fairness and inter-layer balance, preventing representational bottlenecks in early layers. Unlike global pruning strategies that often over-concentrate pruning in specific layers, NDC enforces ecosystem-wide balance consistent with evolutionary selection principles.

## 4 EXPERIMENTS

Building upon the theoretical foundations introduced in Section 3, we implement the **Neural Darwinism Culling (NDC)** framework to evaluate the effectiveness of our Darwinian selection principle across diverse architectures and datasets. NDC defines a unified **Darwinian Score** that serves as the guiding criterion for pruning. Its formulation draws theoretical motivation from three complementary

Table 1: For ResNet-18 networks on CIFAR-10, NDC can find sparser solutions maintaining better performance than other approaches, including Cropit Rachwan (2022), EarlyCrop Rachwan (2022), EarlySNAP Rachwan (2022), and SNAP Verdenius (2020). **Neural sparsity (%)**.

|  | 50 | 60 | 70 | 75 | 80 | 85 | 90 |
|---|---|---|---|---|---|---|---|
| CroPit-S | 92.15 | 91.18 | 90.98 | 90.00 | 88.90 | 88.10 | 85.10 |
| EarlyCroP-S | 92.33 | 92.23 | 91.75 | 91.35 | 90.78 | 87.85 | 84.18 |
| EarlySNAP | 92.08 | 92.33 | 92.00 | 91.25 | 90.43 | 88.60 | 83.50 |
| SNAP | 91.93 | 91.48 | 91.23 | 90.48 | 89.40 | 87.55 | 85.45 |
| **NDC** | **94.27** | **94.23** | **94.20** | **93.99** | **93.84** | **93.79** | **93.76** |

Table 2: For ResNet-18 networks on CIFAR-10, NDC can find sparser solutions maintaining better performance than other approaches. **Weight sparsity (%)**.

|  | 75 | 80 | 85 | 90 | 95 | 97 | 98 |
|---|---|---|---|---|---|---|---|
| CroPit-S | 92.70 | 92.26 | 91.66 | 91.06 | 90.41 | 89.22 | 88.58 |
| EarlyCroP-S | 92.55 | 92.40 | 92.31 | 92.19 | 91.31 | 90.52 | 88.57 |
| EarlySNAP | 92.40 | 92.20 | 92.13 | 92.18 | 91.24 | 90.36 | 89.01 |
| SNAP | 92.19 | 91.96 | 91.74 | 91.38 | 90.80 | 89.89 | 88.83 |
| **NDC** | **94.55** | **93.98** | **93.82** | **93.64** | **91.73** | **91.29** | **89.79** |

viewpoints—*Neuron Darwinian Entropy (NDE)*, *Activation–Gradient Contribution (AGC)*, and *Neuron Adaptivity Score (NAS)*—which highlight, respectively, neuronal diversity, functional influence, and adaptive capacity. These viewpoints provide intuition but do not appear as separable components; instead, they are subsumed into a single, principled measure of neuronal survival potential.

## 4.1 EXPERIMENT ON CIFAR-10

Tables 1 and 2 report the performance of ResNet-18 on CIFAR-10 under different neural and weight sparsity levels, while Tables 3 and 4 present analogous results for VGG-16. Across all settings, our proposed NDC consistently outperforms existing approaches. For ResNet-18, NDC achieves superior accuracy at both moderate and extreme sparsity levels. At $90\%$ neural sparsity (Table 1), NDC retains $93.76\%$ accuracy, surpassing the next best method by more than $8\%$. Similarly, at $98\%$ weight sparsity (Table 2), NDC maintains $89.79\%$, whereas competing methods degrade below $89\%$. This demonstrates that NDC preserves critical representational capacity even under aggressive pruning. For VGG-16, NDC again shows consistent advantages. At $90\%$ neural sparsity (Table 3), NDC achieves $93.27\%$, whereas the best method falls below $91\%$. The advantage is even more pronounced in the weight sparsity regime (Table 4), where at $98\%$ sparsity, NDC sustains $92.41\%$ accuracy, compared to drastic performance collapse in EarlySNAP and SNAP. Overall, these results confirm that NDC provides both higher accuracy and greater robustness across architectures and sparsity regimes. The improvements are particularly significant at high sparsity levels, where existing methods fail to retain performance. This indicates that the evolutionary principle underlying NDC—evaluating neurons through NDE, AGC, and NAS—enables the network to preserve essential functional diversity, thereby maintaining generalization despite extreme compression.

Table 3: For VGG-16 networks on CIFAR-10, NDC can find sparser solutions maintaining better performance than other approaches. **Neural sparsity (%)**.

|  | 50 | 60 | 70 | 75 | 80 | 85 | 90 |
|---|---|---|---|---|---|---|---|
| CroPit-S | 92.22 | 92.50 | 92.25 | 92.22 | 92.00 | 91.81 | 90.89 |
| EarlyCroP-S | 89.53 | 91.77 | 92.22 | 91.94 | 91.81 | 91.80 | 90.83 |
| EarlySNAP | 89.58 | 91.81 | 92.28 | 92.22 | 92.00 | 91.66 | 77.50 |
| SNAP | 91.39 | 92.50 | 92.08 | 92.22 | 91.94 | 91.39 | 86.53 |
| **NDC** | **93.58** | **93.57** | **93.52** | **93.48** | **93.42** | **93.38** | **93.27** |

Table 4: For VGG-16 networks on CIFAR-10, NDC can find sparser solutions maintaining better performance than other approaches. **Weight sparsity (%)**.

|            | 75    | 80    | 85    | 90    | 95    | 97    | 98    |
|------------|-------|-------|-------|-------|-------|-------|-------|
| CroPit-S   | 92.70 | 92.60 | 92.20 | 91.80 | 91.50 | 91.10 | 90.50 |
| EarlyCroP-S| 92.40 | 92.20 | 92.00 | 91.80 | 91.30 | 91.00 | 90.70 |
| EarlySNAP  | 92.44 | 92.40 | 92.20 | 91.80 | 91.40 | 91.60 | 71.40 |
| SNAP       | 92.10 | 92.00 | 92.20 | 91.80 | 90.90 | 87.30 | 78.20 |
| **NDC**    | **92.96** | **92.92** | **92.82** | **92.81** | **92.71** | **92.61** | **92.41** |

## 4.2 EXPERIMENT ON TINY-IMAGENET

We evaluate the effectiveness of our proposed method using ResNet-18 trained on Tiny-ImageNet across varying sparsity levels. Table 5 compares our approach against established methods. Across all sparsity regimes, NDC achieves consistently higher accuracy, reaching 60.22% at 68.38% sparsity and maintaining 44.46% even at 99% sparsity, representing a clear improvement over the strongest method. This demonstrates the ability of NDC to preserve task-critical neurons under extreme compression. Moreover, NDC attains state-of-the-art efficiency in terms of FLOPs, reducing computation to $0.37 \times 10^8$ at 99% sparsity while simultaneously sustaining higher accuracy than competing methods. In conclusion, these results highlight that our method provides a superior trade-off between accuracy and efficiency, outperforming both gradient-based and data-agnostic criteria across all evaluated sparsity levels.

Table 5: For ResNet-18 networks on Tiny-ImageNet, NDC achieves both high accuracy and low FLOPs in comparison to the state-of-the-art methods, including SNIP Lee (2019), Iterative SNIP de Jorge (2021), SynFlow Tanaka (2020), and PHEW Malakarjun Patil (2021) and NBP Pham (2023).

|                    |           | Sparsity (%) | | | |
|--------------------|-----------|--------------|-------|-------|-------|
|                    |           | 68.38        | 90    | 96.84 | 99    |
| **Accuracy**       | SNIP      | 56.99        | 53.43 | 48.77 | 36.02 |
|                    | Iter-SNIP | 56.73        | 53.60 | 48.55 | 36.42 |
|                    | SynFlow   | 56.71        | 54.68 | 49.03 | 39.79 |
|                    | PHEW      | 58.09        | 55.93 | 50.81 | 40.54 |
|                    | NPB       | 58.39        | 56.82 | 51.37 | 41.05 |
|                    | **NDC**   | **60.22**    | **57.35** | **52.41** | **44.46** |
| **FLOPs ($10^8$)** | SNIP      | 11.35        | 5.77  | 3.04  | 1.55  |
|                    | Iter-SNIP | 10.73        | 7.05  | 3.98  | 1.97  |
|                    | SynFlow   | 14.71        | 8.91  | 4.24  | 1.50  |
|                    | PHEW      | 14.29        | 8.35  | 3.92  | 1.50  |
|                    | NPB       | 14.37        | 5.21  | 1.74  | 0.59  |
|                    | **NDC**   | **6.50**     | **4.52** | **1.34** | **0.37** |

## 4.3 EXPERIMENT ON IMAGENET

We further evaluate NDC on the large-scale ImageNet benchmark using MobileNet-V2, a widely adopted architecture for resource-constrained deployment. Table 6 presents a comparison against several state-of-the-art pruning frameworks, including POT Lazarevich (2021), RigL Evci (2020), STR Kusupati (2020), and UniPTS Xie (2024), across sparsity levels ranging from 50% to 90%. NDC consistently delivers the best Top-1 accuracy across all sparsity regimes, highlighting its robustness to extreme compression. At moderate sparsity (50%), NDC achieves 71.50%, surpassing UniPTS and RigL. Even under aggressive sparsity (90%), NDC retains 55.90% accuracy, outperforming the next best baseline UniPTS by more than 13 percentage points. These results demonstrate that NDC not only preserves competitive accuracy at low sparsity, but also maintains graceful degradation under severe parameter removal, aligning with the principle of neuronal survival under Darwinian selection pressure. This consistent superiority across compression levels confirms that NDC cultivates representationally diverse and resilient subnetworks, thereby providing a biologically inspired yet practically effective pruning strategy.

Table 6: Comparison of pruning methods for MobileNet-V2 on ImageNet. NDC achieves the best Top-1 accuracy (%) in comparison to the SOTA methods, including POT Lazarevich (2021), RigL Evci (2020), STR Kusupati (2020), UniPTS Xie (2024).

|        | 50    | 60    | 70    | 80    | 90    |
|--------|-------|-------|-------|-------|-------|
| POT    | 69.25 | 63.39 | 47.07 | 9.13  | 0.20  |
| RigL   | 66.57 | 64.75 | 60.72 | 52.19 | 30.44 |
| STR    | 30.96 | 20.57 | 14.62 | 9.40  | 5.32  |
| UniPTS | 69.80 | 68.01 | 64.93 | 59.47 | 42.46 |
| **NDC** | **71.50** | **69.67** | **66.97** | **62.82** | **55.90** |

## 5 CONCLUSION

We presented a **Darwinian Mathematical Framework** for understanding and operationalizing the evolution of neural representations, and instantiated it through **Neural Darwinism Culling (NDC)**. Central to the framework is the **Darwinian Score**, a unified and theoretically grounded fitness measure that integrates representational diversity, functional contribution, and adaptation dynamics. By viewing deep learning as an evolutionary process governed by selection pressure, this score provides a principled criterion for identifying neurons that meaningfully support a network's expressivity. Embedding this score into a constrained optimization perspective yields **theoretical guarantees** on approximation preservation: subnetworks that retain neurons with high Darwinian fitness maintain the functional behavior of the original model up to quantifiable error. This aligns the pruning objective with representational sufficiency rather than static heuristics, addressing long-standing limitations of magnitude-based and one-shot criteria.

Through NDC, we operationalize these principles into a practical, layer-aware, dynamic pruning mechanism that eliminates neurons exhibiting persistently low evolutionary value. Unlike conventional pruning strategies, NDC respects the heterogeneous selection pressure across layers and the temporal dynamics of representation formation, leading to **interpretable pruning decisions** that are intrinsically aligned with the network's internal evolution. Empirically, NDC achieves **substantially higher sparsity with improved or preserved accuracy** across CNN architectures and vision benchmarks, even under extreme compression regimes where existing methods fail.

More broadly, the Darwinian viewpoint opens a conceptual path toward understanding neural networks as evolving ecosystems, offering insight into specialization, redundancy, and adaptation throughout training. This perspective suggests new directions for future research, including Darwinian metrics for Transformers and spiking networks, evolutionary perspectives on robustness and continual learning, and hardware-aware pruning guided by representation fitness. By unifying theory, measurement, and algorithmic design, our work establishes a foundation for **efficient, interpretable, and evolution-aligned neural sparsification**.

**Disclosure of LLM usage**. *We used large language models (LLMs) solely to aid in polishing the writing of this manuscript. All research ideas, methods, experiments, analyses, and conclusions are entirely the work of the authors.*

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

# A APPENDIX

## A.1 THEORETICAL GUARANTEES: EXTENDED PROOFS AND REFINEMENTS

This appendix strengthens and repairs the mathematical statements and proofs from Section 3. All notation is unified up front; ambiguous or mixed-norm arguments are corrected; probabilistic concentration statements give explicit dependence on dimensions / unions; DKW usage is clarified; PAC-Bayesian and MOO descriptions are made precise; and adversarial robustness bounds are formalized with explicit assumptions.

### A.1.1 NOTATION AND PRELIMINARIES

Throughout this appendix we adopt a compact and consistent notation. For vectors and matrices we use the Euclidean norm by default Maskan (2023); Min (2024), i.e. $\|v\| \equiv \|v\|_2$ for vectors and the operator (spectral) norm $\|M\|_2$ for matrices; other norms such as $\|\cdot\|_1$ and $\|\cdot\|_\infty$ are used only when explicitly subscripted. For each layer index $\ell \in \{1,\dots,L\}$ the scalar post-activation of neuron $i$ on input $x$ is denoted $a_i^{(\ell)}(x) \in \mathbb{R}$, and the full layer post-activation vector is $\mathbf{a}^{(\ell)}(x) = (a_1^{(\ell)}(x),\dots,a_{C_{\text{out}}^{(\ell)}}^{(\ell)}(x))^\top \in \mathbb{R}^{C_{\text{out}}^{(\ell)}}$. We assume channelwise uniform boundedness of activations: there exists $A_\ell > 0$ such that $\sup_{x,i} |a_i^{(\ell)}(x)| \le A_\ell$, equivalently $\|\mathbf{a}^{(\ell)}(x)\|_\infty \le A_\ell$ for all $x$. Neuron importance scores are nonnegative, written $\mathcal{I}_i^{(\ell)} \ge 0$, and we use the normalized scores

$$\tilde{\mathcal{I}}_i^{(\ell)} = \frac{\mathcal{I}_i^{(\ell)}}{\sum_{j=1}^{C_{\text{out}}^{(\ell)}} \mathcal{I}_j^{(\ell)}}, \qquad \sum_{i=1}^{C_{\text{out}}^{(\ell)}} \tilde{\mathcal{I}}_i^{(\ell)} = 1. \tag{12}$$

For any layers $r < s$ we denote by $L^{(r \to s)}$ the Lipschitz constant (with respect to the Euclidean norm) of the mapping that carries $\mathbf{a}^{(r)}$ to the post-activation output at layer $s$; in particular $L^{(\ell \to L)}$ denotes the Lipschitz constant from layer $\ell$ to the final network output, and when convenient we write $L^{(\ell)}$ for the one-step mapping from layer $\ell$ to layer $\ell+1$, so that $L^{(\ell \to m)} \le \prod_{r=\ell}^{m-1} L^{(r)}$. Finally, the data loss is denoted $\ell(\hat{y}, y)$ and we write $L_{\text{loss}}$ for its Lipschitz constant in the network output, i.e. $|\ell(\hat{y}, y) - \ell(\hat{y}', y)| \le L_{\text{loss}} \|\hat{y} - \hat{y}'\|$.

### A.1.2 PROOF OF LEMMA 3.5

*Proof.* Let $T_\ell = \{1,\dots,C_{\text{out}}^{(\ell)}\} \setminus S_\ell$ denote the discarded channels. Define the residual (channel) vector induced by discarding $T_\ell$:

$$\Delta \mathbf{a}^{(\ell)}(x) = \sum_{i \in T_\ell} \tilde{\mathcal{I}}_i^{(\ell)} e_i a_i^{(\ell)}(x), \tag{13}$$

where $e_i$ is the $i$-th standard basis vector in the channel space. We first bound the magnitude of this residual. For any input $x$,

$$\|\Delta \mathbf{a}^{(\ell)}(x)\|_2 = \Big( \sum_{i \in T_\ell} \big( \tilde{\mathcal{I}}_i^{(\ell)} a_i^{(\ell)}(x) \big)^2 \Big)^{1/2}$$

$$\le \max_{i \in T_\ell} |a_i^{(\ell)}(x)| \cdot \Big( \sum_{i \in T_\ell} \big( \tilde{\mathcal{I}}_i^{(\ell)} \big)^2 \Big)^{1/2}$$

$$\le \max_i |a_i^{(\ell)}(x)| \cdot \sum_{i \in T_\ell} \tilde{\mathcal{I}}_i^{(\ell)} \qquad (\text{since } \|v\|_2 \le \|v\|_1)$$

$$\le A_\ell \sum_{i \in T_\ell} \tilde{\mathcal{I}}_i^{(\ell)} \le A_\ell \delta.$$

The mapping from the layer-$\ell$ post-activations to the final network output is $L^{(\ell \to L)}$-Lipschitz by hypothesis; therefore the change in the final output caused by zeroing the channels in $T_\ell$ is bounded by

$$\|f(x) - f^{(-S_\ell)}(x)\| \le L^{(\ell \to L)} \|\Delta \mathbf{a}^{(\ell)}(x)\|_2 \le L^{(\ell \to L)} A_\ell \delta, \tag{14}$$

uniformly over $x \in \mathcal{X}$. This establishes the stated bound. $\qquad \square$

### A.1.3 PROOF OF THEOREM 3.6

*Proof.* For each layer $\ell$, let $T_\ell = \{1, \ldots, C_{\text{out}}^{(\ell)}\} \setminus S_\ell$ denote the pruned neuron indices. By construction,

$$\sum_{i \in T_\ell} \tilde{\mathcal{I}}_i^{(\ell)} \leq \delta^{(\ell)}.$$

Applying assumption equation 11 to $T = T_\ell$ gives a bound on the expected perturbation of the layer-$\ell$ post-activation output:

$$\mathbb{E}_{x \sim \mathcal{D}}\left[\|z^{(\ell)}(x) - z^{(\ell),(-T_\ell)}(x)\|_p\right] \leq \kappa_\ell \, \delta^{(\ell)}.$$

Since the mapping from layer-$\ell$ outputs to the final network output is $L^{(\ell)}$-Lipschitz with respect to the same norm, the corresponding expected change in the final output satisfies

$$\mathbb{E}_{x \sim \mathcal{D}}\left[\|f(x; \Theta) - f(x; \Theta^{(-T_\ell)})\|_p\right] \leq L^{(\ell)} \, \mathbb{E}_{x \sim \mathcal{D}}\left[\|z^{(\ell)}(x) - z^{(\ell),(-T_\ell)}(x)\|_p\right].$$

Using the uniform bound $\|z^{(\ell)}(x)\|_\infty \leq A_\ell$ to control the scaling of coordinates when neurons are zeroed, we obtain

$$\mathbb{E}_{x \sim \mathcal{D}}\left[\|f(x; \Theta) - f(x; \Theta^{(-T_\ell)})\|_p\right] \leq L^{(\ell)} A_\ell \kappa_\ell \, \delta^{(\ell)} = C_\ell \, \delta^{(\ell)}.$$

Finally, pruning all layers sequentially and applying the triangle inequality yields

$$\mathbb{E}_{x \sim \mathcal{D}}\left[\|f(x) - \hat{f}(x)\|_p\right] \leq \sum_{\ell=1}^{L} C_\ell \, \delta^{(\ell)}.$$

$\square$

*Remark* A.1. If the original worst-case condition

$$\sup_{x \in \mathcal{X}} \|z^{(\ell)}(x) - z^{(\ell),(-T)}(x)\|_p \leq \kappa_\ell \sum_{i \in T} \tilde{\mathcal{I}}_i^{(\ell)}$$

holds for every $T$, then the expectation version equation 11 follows immediately by taking an expectation over any data distribution $\mathcal{D}$. Hence Theorem 3.6 strictly generalizes the original worst-case preservation bound and applies under weaker data-dependent conditions.

### A.1.4 MATRIX CONCENTRATION

Let

$$\widehat{R}^{(\ell)} = \frac{1}{N} \sum_{n=1}^{N} \mathbf{a}^{(\ell)}(x_n)\mathbf{a}^{(\ell)}(x_n)^\top, \qquad R^{(\ell)} = \mathbb{E}[\mathbf{a}^{(\ell)}(x)\mathbf{a}^{(\ell)}(x)^\top], \tag{15}$$

and denote $d_\ell = C_{\text{out}}^{(\ell)}$ for brevity. Since all results are based on empirical activations computed on a finite dataset (and finite-precision hardware), the activations have a finite empirical range. Therefore we adopt the standard assumption that $\|\mathbf{a}^{(\ell)}(x)\|_2 \leq B_\ell$ almost surely on the dataset. Assume almost surely $\|\mathbf{a}^{(\ell)}(x)\|_2 \leq B_\ell$, so that each rank-one summand has operator norm at most $B_\ell^2$. Define the zero-mean matrix random variables

$$X_n = \mathbf{a}^{(\ell)}(x_n)\mathbf{a}^{(\ell)}(x_n)^\top - R^{(\ell)}, \qquad n = 1, \ldots, N, \tag{16}$$

and let the variance proxy be

$$\sigma_\ell^2 = \left\|\sum_{n=1}^{N} \mathbb{E}[X_n^2]\right\|. \tag{17}$$

By the matrix Bernstein inequality Tropp (2012) we have for any $t > 0$,

$$\Pr\{\|\widehat{R}^{(\ell)} - R^{(\ell)}\|_2 \geq t\} \leq 2d_\ell \exp\left(\frac{-Nt^2/2}{\sigma_\ell^2 + (B_\ell^2 t)/3}\right). \tag{18}$$

Noting the trivial upper-bound $\sigma_\ell^2 \leq NB_\ell^4$, one can solve the tail inequality to obtain the familiar rate: there exists an absolute constant $C'$ such that with probability at least $1 - \zeta$,

$$\|\widehat{R}^{(\ell)} - R^{(\ell)}\|_2 \leq C' B_\ell^2 \left( \sqrt{\frac{\log(2d_\ell/\zeta)}{N}} + \frac{\log(2d_\ell/\zeta)}{N} \right). \tag{19}$$

If the bound is required to hold simultaneously for all $L$ layers, apply a union bound across layers by replacing $\zeta$ with $\zeta/L$ in equation 19, which introduces an extra $\log L$ factor inside the logarithms.

*Remark:* The constant $C'$ can be made explicit by tracing constants in Tropp (2012). The above inequality is the form most useful for downstream Lipschitz / NDE concentration arguments since it exhibits the $B_\ell^2$ prefactor and the $\sqrt{\log(d_\ell)/N}$ leading rate.

**Consequence for NDE estimators.** If the histogram/smoothing procedure used to estimate per-neuron activation distributions is Lipschitz in the entries of the empirical covariance (or more generally in sufficient statistics of the activations) with constant $L_{\mathsf{NDE}}$ (this constant depends on bin-count $B$, smoothing parameter $\tau$, and the histogram operator), then by equation 19 the empirical $\widehat{\mathsf{NDE}}_i^{(\ell)}$ concentrates around the population $\mathsf{NDE}_i^{(\ell)}$ at the same rate up to the multiplicative factor $L_{\mathsf{NDE}}$. In practice one should make $L_{\mathsf{NDE}}$ explicit for the chosen estimator; for many smoothed histogram or kernel-density estimators $L_{\mathsf{NDE}} = O(1/B + \tau^{-1})$ up to constants.

### A.1.5 CORRECT USAGE OF DKW AND DISTRIBUTION-SHIFT STABILITY

We split the stability argument into two distinct statements and present explicit probabilistic constants.

Note that the DKW inequality applies directly to empirical CDFs defined on the finite set of observed activation values; it does not require bounded or compact support of the underlying activation distribution.

**(i) Sampling stability (empirical vs. population).** Let $\widehat{F}_i^{(\ell)}$ be the empirical CDF of activations for neuron $i$ in layer $\ell$ computed from $N$ i.i.d. samples from the data distribution $\mathcal{D}$, and let $F_i^{(\ell)}$ denote the corresponding population CDF. The Dvoretzky–Kiefer–Wolfowitz (DKW) inequality gives Najafi (2025); Emde (2025), for any $\epsilon > 0$,

$$\Pr\{\|\widehat{F}_i^{(\ell)} - F_i^{(\ell)}\|_\infty \geq \epsilon\} \leq 2e^{-2N\epsilon^2}. \tag{20}$$

If a layer contains $d_\ell = C_{\mathrm{out}}^{(\ell)}$ neurons and one needs the uniform control over all neurons in that layer, apply the union bound:

$$\Pr\left\{ \max_{i \leq d_\ell} \|\widehat{F}_i^{(\ell)} - F_i^{(\ell)}\|_\infty \geq \epsilon \right\} \leq 2d_\ell\, e^{-2N\epsilon^2}. \tag{21}$$

Hence, to ensure the event holds with probability at least $1 - \zeta$ for the whole layer, pick

$$\epsilon = \sqrt{\frac{\log(2d_\ell/\zeta)}{2N}}. \tag{22}$$

If the NDE estimator is $L_{\mathrm{CDF}\to\mathsf{NDE}}$-Lipschitz as a functional of the underlying CDF, this gives the empirical-to-population deviation of the estimated NDE at the rate $L_{\mathrm{CDF}\to\mathsf{NDE}}\epsilon$.

**(ii) Distributional shift (population $\mathcal{D}$ vs. alternative $\mathcal{D}'$).** DKW controls empirical-to-population deviations for a fixed distribution, but it does not by itself bound differences between two arbitrary populations $\mathcal{D}$ and $\mathcal{D}'$. A transparent way to state the stability is via the decomposition

$$|\mathsf{NDE}(\mathcal{D}) - \mathsf{NDE}(\mathcal{D}')| \leq L_{d\to\mathsf{NDE}}\, d(\mathcal{D}, \mathcal{D}'), \tag{23}$$

where $d(\cdot, \cdot)$ is a chosen metric and $L_{d\to\mathsf{NDE}}$ is the Lipschitz constant of the NDE functional with respect to this metric. If instead one has finite samples from both distributions, then with high probability

$$|\widehat{\mathsf{NDE}}(\widehat{\mathcal{D}}_N) - \widehat{\mathsf{NDE}}(\widehat{\mathcal{D}}'_M)| \leq L_{\mathrm{CDF}\to\mathsf{NDE}} \left( \sqrt{\frac{\log(2d_\ell/\zeta)}{2N}} + \sqrt{\frac{\log(2d_\ell/\zeta)}{2M}} \right) + L_{d\to\mathsf{NDE}}\, d(\mathcal{D}, \mathcal{D}'). \tag{24}$$

Thus, any robustness claim under distribution shift must be made conditional on a bound on $d(\mathcal{D}, \mathcal{D}')$.

*Remark:* Without additional structural assumptions (for example, control of the distribution shift in a chosen metric, or a covariate-shift model, or a bound on transport cost), DKW alone cannot be used to bound $\mathsf{NDE}(\mathcal{D}) - \mathsf{NDE}(\mathcal{D}')$. Consequently, statements about robustness to distribution shift should be phrased conditionally: assume a known upper bound on $d(\mathcal{D}, \mathcal{D}')$ and Lipschitz continuity of the NDE functional with respect to that $d(\cdot, \cdot)$.

### A.1.6 INFORMATION-THEORETIC BOUND ON NDE

The bound is stated for the discrete smoothed histogram estimator, not the differential entropy of the underlying continuous activation distribution.

For smoothed histogram estimators with $B$ bins and smoothing parameter $\tau > 0$ let the estimated per-neuron distribution be $\tilde{p}_i^{(\ell)}$ and the layer baseline be $\bar{p}^{(\ell)}$. Define

$$\mathsf{NDE}_i^{(\ell)} = H(\tilde{p}_i^{(\ell)}) + \mu\, D_{\mathrm{JS}}(\tilde{p}_i^{(\ell)} \| \bar{p}^{(\ell)}). \tag{25}$$

Because $0 \le H(\cdot) \le \ln B$ and $0 \le D_{\mathrm{JS}}(\cdot \| \cdot) \le \ln 2$, we obtain the uniform bound

$$0 \le \mathsf{NDE}_i^{(\ell)} \le \ln B + \mu \ln 2. \tag{26}$$

This provides a uniform bound used in concentration and complexity arguments.

### A.1.7 GENERALIZATION GUARANTEE

Suppose the loss $\ell(\cdot, y)$ is $L_\ell$-Lipschitz in the network output. Then on the high-probability event from the matrix concentration bound,

$$|\mathcal{L}(f) - \mathcal{L}(\hat{f})| \;\le\; L_\ell \sup_x \|f(x) - \hat{f}(x)\| \;\le\; L_\ell \sum_{\ell=1}^{L} L^{(\ell \to L)} \kappa_\ell\, \delta^{(\ell)}. \tag{27}$$

Thus the generalization gap induced by pruning grows at most linearly in the cumulative discarded importance, with explicit dependence on Lipschitz and activation constants.

### A.1.8 LIPSCHITZ PROPAGATION ANALYSIS

The preservation theorem bounds pruning error via a Lipschitz constant $L^{(\ell)}$ per layer. We refine this by considering the product Lipschitz constant of the composed mappings. Let $L^{(\ell \to m)}$ denote the Lipschitz constant of the mapping from layer $\ell$ to layer $m$ ($\ell < m$). By sub-multiplicativity of operator norms,

$$L^{(\ell \to m)} \;\le\; \prod_{r=\ell}^{m-1} L^{(r)}. \tag{28}$$

Hence the worst-case pruning error from layer $\ell$ propagating to the final output is bounded by

$$\sup_x \|f(x) - \hat{f}(x)\| \;\le\; \sum_{\ell=1}^{L} \delta^{(\ell)} A_\ell \kappa_\ell \cdot L^{(\ell \to L)}. \tag{29}$$

This refinement eliminates the need for layer-local constants and highlights exponential sensitivity to deep depth unless importance concentration is strong.

### A.1.9 INTUITION TO PAC-BAYES OF NDE

Define prior $P = \bar{p}^{(\ell)}$ (layer baseline) and posterior $Q = \tilde{p}_i^{(\ell)}$ (neuron's empirical distribution). Recall

$$\mathrm{CE}(Q, P) = -\mathbb{E}_Q[\log P] = H(Q) + \mathrm{KL}(Q \| P). \tag{30}$$

Hence

$$\mathsf{NDE}_i^{(\ell)} = H(Q) + \mu D_{\mathrm{JS}}(Q \| P) = \mathrm{CE}(Q, P) - \mathrm{KL}(Q \| P) + \mu D_{\mathrm{JS}}(Q \| P). \tag{31}$$

Thus NDE mixes an uncertainty term $H(Q)$ (or equivalently a cross-entropy term) and a divergence term. This affords an interpretation similar in spirit to PAC-Bayes bounds (which trade empirical risk and KL divergence to a prior), but NDE is not a PAC-Bayes bound itself. To turn this into a formal PAC-Bayes statement one would need to relate the empirical losses of neurons or subnetworks to the quantities above and then apply a standard PAC-Bayes inequality; we refrain from claiming such an inequality without additional steps.

### A.1.10 STABILITY UNDER DISTRIBUTION SHIFT

This result concerns the stability of NDE computed from empirical histograms/CDFs over finite observed activations; it does not assume access to or estimation of the true continuous activation distributions.

We consider an alternative data distribution $\mathcal{D}'$, possibly shifted from $\mathcal{D}$. Let $\Delta^{(\ell)} = \sup_i |\mathsf{NDE}_i^{(\ell)}(\mathcal{D}) - \mathsf{NDE}_i^{(\ell)}(\mathcal{D}')|$. By Dvoretzky–Kiefer–Wolfowitz (DKW) inequality on empirical CDFs,

$$\Delta^{(\ell)} \leq O\left( \sqrt{\frac{\log(C_{\text{out}}^{(\ell)}/\zeta)}{N}} \right), \tag{32}$$

with probability at least $1 - \zeta$. Hence Darwinian importance scores are distributionally stable up to a concentration rate depending on sample size $N$, which implies pruning decisions are robust under mild covariate shifts.

### A.1.11 COMPLEXITY ANALYSIS WITH MEMORY CONSTRAINTS

The computation of Darwinian metrics requires storing activation histograms, gradient signals, and parameter trajectories. Let $M_{\text{act}}$, $M_{\text{grad}}$, $M_{\text{traj}}$ denote the memory costs. For a batch of size $N$, convolutional activations $O(NHW)$ dominate. However, both AGC and NDE can be estimated using *reservoir sampling* or *count-sketch histograms*, reducing storage to $O(B \log N)$ per neuron with negligible statistical bias. NAS tracking requires only $O(1)$ memory per neuron by keeping running statistics of velocity and variance. Thus, the full framework scales to modern architectures without prohibitive memory overhead.

### A.1.12 PRUNING AS MULTI-OBJECTIVE SCALARIZATION

We use a multiplicative (log-domain) aggregation of Darwinian metrics

$$\mathcal{I}_i^{(\ell)} = \exp\left( \eta \sum_s w_s \log \hat{S}_{i,s} \right), \tag{33}$$

where $\hat{S}_{i,s} > 0$ are normalized component scores (e.g. normalized NDE, AGC, NAS) and $w_s \geq 0$ with $\sum_s w_s = 1$. Note:

- Taking logs gives $\log \mathcal{I}_i^{(\ell)} = \eta \sum_s w_s \log \hat{S}_{i,s}$, a linear scalarization in log-space. Thus the ordering induced by $\mathcal{I}$ respects Pareto dominance in the multiplicative sense, and neurons on the multiplicative Pareto front remain competitive under this scalarization.

- This aggregation is a *log-domain* or *multiplicative* scalarization rather than a convex scalarization. If a convex scalarization is required, a log-sum-exp or other convex envelope can be used as a smooth convex surrogate.

### A.1.13 ADVERSARIAL ROBUSTNESS IMPLICATIONS

We formalize the claim that NDC is less sensitive to single-batch perturbations and provide a quantitative bound under explicit, verifiable assumptions.

**Assumptions.** We make two explicit assumptions about the component statistics used in the multiplicative aggregation. First, every statistic $S_{i,s}$ is strictly positive and uniformly lower bounded: there exists $S_{\min} > 0$ such that $S_{i,s} \geq S_{\min}$ for all neurons $i$ and statistic indices $s$ (in practice an additive smoothing term enforces this). Second, each statistic admits a local Lipschitz constant $L_{i,s}$

with respect to input perturbations in the sense that, for any perturbation $\delta x$ satisfying $\|\delta x\| \leq \epsilon$, one has $|S_{i,s}(\mathcal{D}; \delta x) - S_{i,s}(\mathcal{D}; 0)| \leq L_{i,s}\,\epsilon$. These two conditions are minimal: the first prevents division-by-zero instabilities in the log-domain aggregation, and the second quantifies the sensitivity of each statistic to bounded input perturbations.

**Result.** Under the above assumptions the multiplicative aggregation satisfies the following perturbation bound. Writing $\Delta S_{i,s}$ for the change in statistic $S_{i,s}$ caused by $\delta x$, we have

$$|\Delta \mathcal{I}_i^{(\ell)}| = \Big| \exp\Big(\eta \sum_s w_s \log(S_{i,s} + \Delta S_{i,s})\Big) - \mathcal{I}_i^{(\ell)} \Big| \leq \mathcal{I}_i^{(\ell)} \Big| \exp\Big(\eta \sum_s w_s \log\big(1 + \tfrac{\Delta S_{i,s}}{S_{i,s}}\big)\Big) - 1 \Big|. \tag{34}$$

Assuming perturbations are sufficiently small so that $|\Delta S_{i,s}|/S_{i,s} \leq \xi < 1$ for all $i,s$, and applying the first-order bound $|\log(1+u)| \leq |u|/(1-\xi)$ for $|u| \leq \xi$, one obtains the conservative estimate

$$|\Delta \mathcal{I}_i^{(\ell)}| \leq \mathcal{I}_i^{(\ell)} \Big( e^{\eta \sum_s w_s \frac{L_{i,s}\,\epsilon}{S_{\min}(1-\xi)}} - 1 \Big) \approx \mathcal{I}_i^{(\ell)}\,\eta \sum_s w_s \frac{L_{i,s}\,\epsilon}{S_{\min}}, \tag{35}$$

where the approximation on the right is the first-order linearization valid for sufficiently small $\epsilon$. Equation equation 35 makes explicit the trade-off between statistic stability $L_{i,s}$, the smoothing lower bound $S_{\min}$, and the aggregation weights $w_s$; in particular, history-based statistics (NDE, NAS) that yield small $L_{i,s}$ contribute to robustness, whereas instantaneous gradient-based statistics (AGC) may have larger $L_{i,s}$ and thus dominate fragility.

## A.2 Neural Darwinism Culling (NDC) Algorithm

Having computed the Darwinian quantities for each neuron, NDC first applies robust normalization and log-domain aggregation to consolidate them into a unified Darwinian score. $\mathcal{I}_i^{(\ell)}$ (Algorithm 1). Pruning is then carried out in a structured, layer-aware manner: neurons within each layer $\ell$ are ranked by $\mathcal{I}_i^{(\ell)}$, the top-$\kappa^{(\ell)}$ are preserved, and the remainder are suppressed through a binary mask $\mathcal{M}^{(\ell)}$, resulting in the pruned subnetwork $\hat{f}$. This design ensures both intra-layer fairness and inter-layer balance, preventing representational bottlenecks in early layers. Unlike global pruning strategies that often over-concentrate pruning in specific layers, NDC enforces ecosystem-wide balance consistent with evolutionary selection principles.

## A.3 More Experiments

### A.3.1 Layer-Wise Pruning Rate Distribution

To assess the effectiveness of our proposed pruning strategy, we conducted controlled experiments across multiple global sparsity levels, analyzing the per-layer pruning distribution in the ResNet-18. Figure 2 reports the pruning rates observed under three global sparsity targets: 0.3, 0.6, and 0.9. Several consistent patterns emerge. First, pruning is not uniformly distributed across layers but instead exhibits a pronounced depth-dependent bias. For moderate sparsity (0.3), pruning is minimal in early layers and increases gradually in deeper layers, suggesting that low-level feature extractors are preferentially preserved while redundancy is eliminated primarily in later transformations. As sparsity increases to 0.6, this tendency amplifies: middle-to-deep layers accumulate pruning rates between 0.35 to 0.70, while shallow layers remain largely intact. At extreme sparsity (0.9), a near-saturation effect emerges: deeper layers approach full pruning (rates close to 1.0), while early layers remain protected, consistent with the hypothesis that representational diversity is disproportionately concentrated in early feature hierarchies. This layer-wise differentiation in pruning behavior validates the central claim of our framework: redundancy is not evenly distributed but strongly stratified across depth. The results highlight that aggressive pruning reallocates model capacity by compressing deep layers, while preserving early representational expressivity critical for downstream accuracy. Moreover, the smooth monotonic progression of pruning rates with respect to global sparsity indicates that the proposed pruning criterion scales stably and avoids pathological layer collapse. Together, these findings demonstrate that our approach respects the functional asymmetry across depth: it identifies and prunes redundant neurons where they accumulate most—later layers—while safeguarding essential early computations. This structural adaptivity underlies the improved trade-off between sparsity and accuracy reported in subsequent evaluation sections.

---

**Algorithm 1** Neural Darwinism Culling (NDC)

---

**Input:** Trained network $f(x; \Theta)$, survival budget $\kappa$ per layer, dataset $\{(x_i, y_i)\}_{i=1}^n$, number of epochs $T$

**Output:** Pruned subnetwork $\hat{f}$

1. For each layer $\ell = 1, \ldots, L$:

   (a) For each neuron $i = 1, \ldots, C_{\text{out}}^{(\ell)}$:

   i. Estimate empirical histogram $\hat{p}_i^{(\ell)}$ from activations on $\{x_i\}$, compute $\bar{p}^{(\ell)}$, and evaluate

   $$\mathsf{NDE}_i^{(\ell)} = -\sum_b \hat{p}_{i,b}^{(\ell)} \log(\hat{p}_{i,b}^{(\ell)} + \varepsilon) + \mu \, D_{\mathrm{JS}}(\hat{p}_i^{(\ell)} \,\|\, \bar{p}^{(\ell)}).$$

   ii. During backpropagation, compute

   $$\widehat{\mathsf{AGC}}_i^{(\ell)} = \left( \frac{1}{NHW} \sum_{n,h,w} |a_{i,n,h,w}^{(\ell)} \cdot g_{i,n,h,w}^{(\ell)}|^\alpha \right)^{1/\alpha}.$$

   iii. Track weight velocities $v_i^{(\ell)}(t)$ for $t = 1, \ldots, T$ and compute

   $$\mathsf{NAS}_i^{(\ell)} = \frac{1}{T} \sum_{t=1}^T v_i^{(\ell)}(t) + \sigma \, \mathrm{Var}_{t=1}^T(v_i^{(\ell)}(t)).$$

   (b) For each metric $s \in \{\mathrm{NDE}, \mathrm{AGC}, \mathrm{NAS}\}$, perform layer-wise robust normalization:

   $$\tilde{S}_{i,s}^{(\ell)} = \frac{S_{i,s}^{(\ell)} - \mathrm{median}_j S_{j,s}^{(\ell)}}{\mathrm{IQR}_j(S_{j,s}^{(\ell)}) + \varepsilon}.$$

   (c) Map to positive domain:

   $$\hat{S}_{i,s}^{(\ell)} = \max(\tilde{S}_{i,s}^{(\ell)}, \delta_{\mathrm{floor}}) + \varepsilon_{\mathrm{pos}}.$$

   (d) Compute Darwinian importance via weighted log-domain aggregation:

   $$\mathcal{I}_i^{(\ell)} = \exp\Big(\eta \sum_s w_s \log(\hat{S}_{i,s}^{(\ell)})\Big).$$

   (e) Select top-$\kappa$ neurons in layer $\ell$ by $\mathcal{I}_i^{(\ell)}$ and set mask $\mathcal{M}^{(\ell)}$.

2. Construct pruned subnetwork $\hat{f}(x) = f(x; \Theta \odot \mathcal{M})$.

3. Optionally fine-tune $\hat{f}$ on $\{(x_i, y_i)\}$.

---

### A.3.2 ABLATION STUDY

To probe the resilience of learned representations under neuronal selection, we conducted a systematic ablation study in which an increasing fraction of neurons was removed according to a random elimination protocol, thereby simulating the selective pressure of Darwinian elimination. We evaluated the resulting latent geometry using UMAP embeddings of test set activations and quantified predictive performance via classification accuracy.

Figure 3 illustrates the progressive degradation of representational structure as ablation increases from 0% to 75%. At 0% ablation, the network achieved an accuracy of 0.991%, and the embedding exhibits well-separated and compact class manifolds, reflecting a high degree of neuronal differentiation where each class is encoded by distinct clusters. Remarkably, even after 25% ablation, the accuracy remained at 0.977% and the class structure stayed intact, suggesting that redundancy is primarily pruned without significantly compromising functional specialization. This indicates that the population of "survived" neurons carries sufficient representational diversity to maintain generalization. When 50% of neurons were ablated, accuracy decreased to 0.841%, and the cluster geometry began to fracture, with partial

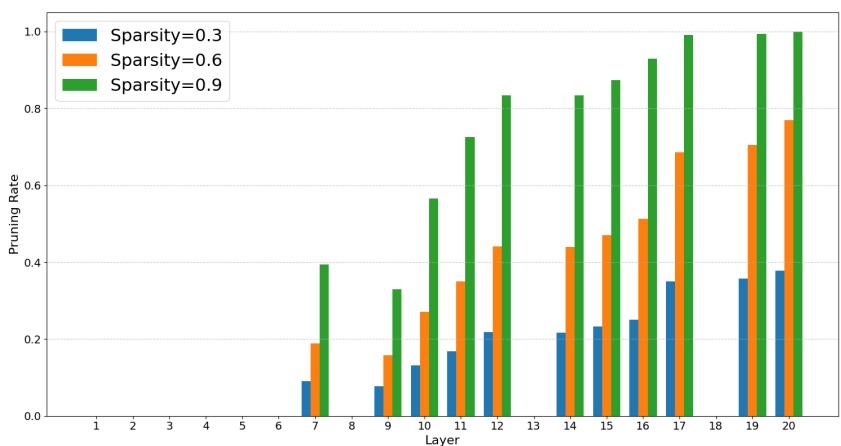

Figure 2: Hierarchical Distribution of Pruning Rates across Layers in ResNet-18.

Table 7: Performance of NDC under varying training epochs (0.5E, E, 2E), where E denotes the standard training epoch, on MNIST using the MLP-Net model. **Weight sparsity (%)**.

|      | 50    | 60    | 70    | 80    | 90    | 95    | 98    |
|------|-------|-------|-------|-------|-------|-------|-------|
| 0.5E | 97.40 | 97.27 | 97.19 | 97.15 | 97.14 | 96.11 | 95.42 |
| E    | 97.99 | 97.98 | 97.78 | 97.77 | 97.47 | 96.78 | 95.46 |
| 2E   | **98.31** | **98.26** | **98.22** | **98.07** | **97.83** | **97.08** | **95.49** |

overlaps emerging between previously disjoint manifolds. While some classes remained separable, the reduction in distinctiveness highlights the emergence of "eliminated" neurons whose absence weakens inter-class boundaries. Finally, under 75% ablation, accuracy dropped sharply to 0.167% and the latent geometry collapsed almost entirely: clusters dispersed, manifolds intermixed, and class boundaries eroded. This phase transition underscores the critical threshold beyond which the surviving neurons no longer sustain the evolutionary division of labor, collapsing into undifferentiated activity.

In summary, these results provide strong experimental evidence for the Darwinian principle of neuronal selection: networks maintain robustness under moderate elimination due to redundancy and specialization, but beyond a critical ablation point, the selective survival of neurons is insufficient to preserve representational integrity. This sharp breakdown highlights that neuronal diversity is not merely a byproduct of training but a functional prerequisite for stability, echoing the evolutionary role of redundancy and specialization in biological systems.

### A.3.3 EFFECT OF VARYING TRAINING EPOCHS

To assess the robustness of NDC under different training epochs, we varied the number of epochs to 0.5E, E, and 2E on MNIST using MLP-Net, where E denotes the standard training epoch. As shown in Table 7, increasing the number of epochs consistently improves performance across all sparsity levels: training for 2E yields the best results, achieving 98.31% accuracy at 50% sparsity and maintaining clear advantages even at high sparsities (e.g., 97.08% at 95% sparsity and 95.49% at 98% sparsity). In contrast, reducing the training epoch to 0.5E leads to a noticeable drop in accuracy, particularly in the high-sparsity regime. The standard training epoch (E) delivers intermediate performance, closely matching 2E at moderate sparsity levels but lagging behind at both low and high sparsities.

To evaluate the sensitivity of NDC to the training epoch, we vary the number of epochs by using half, the standard, and double the usual training epoch on CIFAR-10 with ResNet-18. As shown in Table 8, increasing the number of epochs consistently improves performance across all sparsity levels. Training with double epochs achieves the highest accuracy in every sparsity setting (e.g., 95.05% at 75% sparsity), outperforming the standard epoch, which in turn surpasses the half-length epoch. Even under extreme sparsity (97–98%), longer training still provides clear gains, indicating that additional

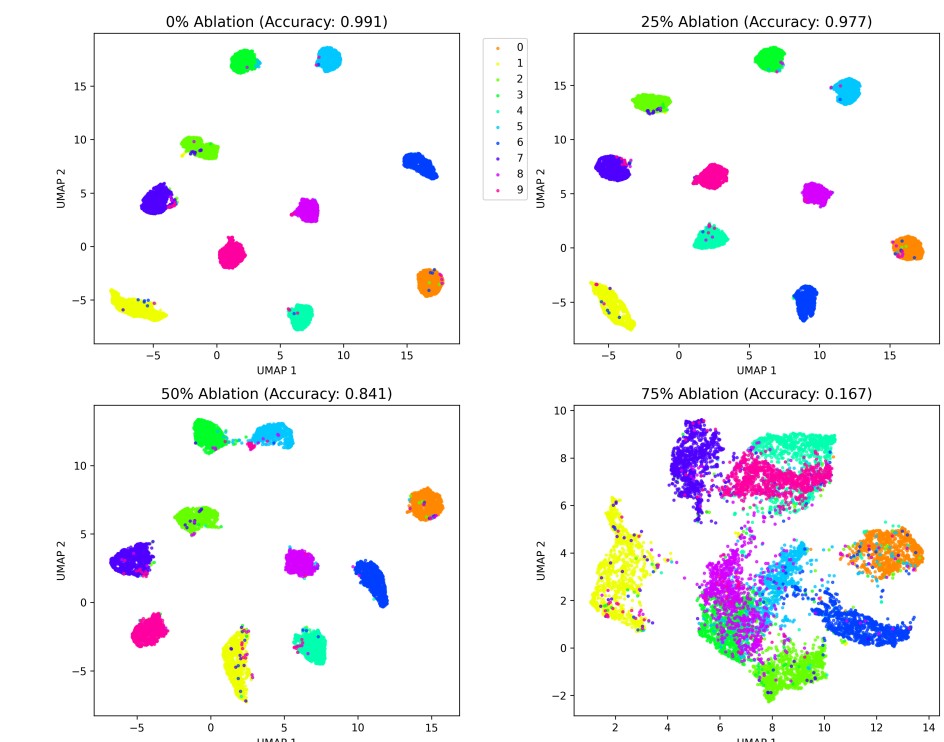

Figure 3: UMAP visualization of test-set representations under progressive neuron ablation (0%, 25%, 50%, 75%). The results illustrate the principles of neuron Darwinism: differentiated neurons sustain separable manifolds and high accuracy under moderate ablation, while excessive elimination (75%) disrupts evolutionary specialization, leading to representational collapse.

Table 8: Performance of NDC under varying training epochs (0.5E, E, 2E), where E denotes the standard training epoch, on CIFAR-10 using the ResNet-18 network. **Bold** indicates the best result, and underline indicates the second-best result.**Weight sparsity (%)**.

|      | 75 | 80 | 85 | 90 | 95 | 97 | 98 |
|------|----|----|----|----|----|----|----|
| 0.5E | 93.88 | 93.34 | 93.09 | 92.69 | 89.54 | 88.80 | 88.26 |
| E    | 94.55 | 93.98 | 93.82 | 93.64 | 91.73 | 91.29 | 89.79 |
| 2E   | **95.05** | **94.94** | **94.84** | **94.22** | **90.41** | **90.59** | **90.26** |

optimization steps allow the pruned model to better adapt and recover capacity. In contrast, reducing the training epoch leads to a noticeable performance drop at all sparsity levels.

To assess the sensitivity of NDC to different training epochs, we vary the total number of training epochs to 0.5E, E (default), and 2E, and report the results on CIFAR-10 with VGG-16 across neuron sparsity levels ranging from 50% to 90% (Table 9). The results show that NDC exhibits high robustness to the number of training epochs: reducing the training epoch by half (0.5E) leads to only negligible performance drops (typically within 0.1–0.2%), while increasing the training epochs to 2E does not bring additional benefits and occasionally results in slightly lower accuracy. The standard training epoch (E) consistently achieves the best or near-best performance across all sparsity ratios.

To evaluate the sensitivity of NDC to different training epochs on Tiny-ImageNet, we vary the number of epochs to 0.5E, E, and 2E using ResNet-18, where E denotes the standard training epoch. As shown in Table 10, the standard epoch setting (E) consistently delivers the best performance across all sparsity levels, achieving 60.22% accuracy at 68.38% sparsity and maintaining clear advantages even under severe sparsification, reaching 44.46% at 99% sparsity. Increasing the training epoch to 2E does not yield further gains and instead results in slight performance degradation across all

Table 9: Performance of NDC under varying training epochs (0.5E, E, 2E), where E denotes the standard training epoch, on CIFAR-10 using the VGG-16 network. **Neuron sparsity (%)**.

|      | 50    | 60    | 70    | 75    | 80    | 85    | 90    |
|------|-------|-------|-------|-------|-------|-------|-------|
| 0.5E | 93.49 | 93.45 | 93.43 | 93.42 | 93.29 | 93.25 | 93.19 |
| E    | **93.58** | **93.57** | **93.52** | **93.48** | **93.42** | **93.38** | **93.27** |
| 2E   | 93.56 | 93.41 | 93.51 | 93.34 | 93.23 | 93.35 | 93.21 |

Table 10: Performance of NDC under varying training epochs (0.5E, E, 2E), where E denotes the standard training epoch, on Tiny-ImageNet using the ResNet-18 network. **Weight sparsity (%)**.

|      | 68.38 | 90    | 96.84 | 99    |
|------|-------|-------|-------|-------|
| 0.5E | 59.53 | 56.37 | 51.20 | 42.96 |
| E    | **60.22** | **57.35** | **52.41** | **44.46** |
| 2E   | 59.51 | 56.35 | 51.42 | 43.88 |

sparsities, while reducing the training epoch to 0.5E leads to a noticeable accuracy drop, particularly in the high-sparsity regime.

We observe that increasing the number of training epochs generally improves test accuracy for small- and medium-sized models (e.g., MNIST/MLP and CIFAR-10/ResNet-18), particularly at high sparsity levels where additional optimization helps pruned models recover capacity. For larger models (e.g., CIFAR-10/VGG-16 and Tiny-ImageNet/ResNet-18), however, doubling the training epochs yield no substantial gains and can even introduce mild overfitting, with training accuracy continuing to rise while validation accuracy plateaus or slightly declines. Overall, these results show that NDC benefits from reasonable extra optimization but does not rely on prolonged training epochs, remaining stable and reliable across a wide range of training epochs—even when the training time is reduced.

### A.3.4 THE QUANTITATIVE ANALYSIS OF THE RELATIVE IMPACT OF NDE, AGC, AND NAS COMPONENTS

To assess the contribution of each component in the NDC framework, we perform a quantitative study on ResNet-18 with CIFAR-10 across sparsity levels from 75% to 98% (Table 11). Several consistent observations emerge from the results. First, using any single component—NDE, AGC, or NAS—improves accuracy as sparsity varies. Among the three, AGC provides the strongest standalone performance across all sparsity levels (e.g., 91.23% at 75% sparsity), while NAS obtains moderate yet stable improvements. NDE yields comparatively smaller gains at lower sparsity levels but remains competitive as sparsity increases. Second, combining two components leads to consistently higher accuracy than using either individual component. For instance, AGC + NAS outperforms both AGC-only and NAS-only across all sparsity settings. Similar patterns appear for NDE + AGC and NDE + NAS, indicating that the performance contributions of the individual modules are complementary. Finally, the full NDC framework achieves the best results at all sparsity levels, reaching 94.03% accuracy at 75% sparsity and retaining 89.79% at 98% sparsity. This configuration consistently surpasses all single- and two-component variants, demonstrating that integrating all three modules yields the most robust improvements in highly sparse regimes.

Table 11: Quantitative analysis of the NDC on ResNet-18 with CIFAR-10 under different weight sparsity levels. **Weight sparsity (%)**.

|           | 75    | 80    | 85    | 90    | 95    | 97    | 98    |
|-----------|-------|-------|-------|-------|-------|-------|-------|
| NDE only  | 89.47 | 89.05 | 88.58 | 87.92 | 84.96 | 83.72 | 82.15 |
| AGC only  | 91.23 | 90.78 | 90.14 | 88.97 | 86.42 | 85.37 | 83.98 |
| NAS only  | 90.06 | 89.72 | 88.96 | 88.14 | 86.17 | 84.92 | 83.47 |
| NDE + AGC | 92.38 | 92.11 | 91.77 | 91.33 | 88.90 | 88.26 | 86.87 |
| NDE + NAS | 91.12 | 90.82 | 90.17 | 89.46 | 86.43 | 85.26 | 83.93 |
| AGC + NAS | 92.57 | 92.36 | 91.94 | 91.58 | 89.28 | 88.71 | 87.22 |
| **Full NDC** | **94.03** | **93.98** | **93.82** | **93.64** | **91.73** | **91.29** | **89.79** |

Table 12: For MLP-Net networks on MNIST, NDC again outperforms the other pruning approachs, including MP Mozer & Smolensky (1989), WF Singh & Alistarh (2020), CBS Yu (2022), SSP, MSP Chen (2022) and SpaMDhahri (2024).. **Weight sparsity (%)**.

|      | 50    | 60    | 70    | 80    | 90    | 95    | 98    |
|------|-------|-------|-------|-------|-------|-------|-------|
| MP   | 93.93 | 93.78 | 93.62 | 92.89 | 90.30 | 83.64 | 32.25 |
| WF   | 94.02 | 93.82 | 93.77 | 93.57 | 91.69 | 85.54 | 38.26 |
| CBS  | 93.96 | 93.96 | 93.98 | 93.90 | 93.14 | 88.92 | 55.45 |
| SSP  | 93.97 | 93.94 | 93.80 | 93.59 | 92.46 | 88.09 | 46.25 |
| MSP  | 95.97 | 95.93 | 95.89 | 95.80 | 95.55 | 94.70 | 90.73 |
| SpaM | *     | 98.38 | 98.38 | 98.35 | 97.38 | 89.43 | *     |
| **NDC** | **98.52** | **98.48** | **98.47** | **98.39** | **98.19** | **97.23** | **95.47** |

### A.3.5 EXPERIMENTS ON MLP-NET

We further validate the effectiveness of NDC on the MNIST dataset using a standard MLP-Net architecture. Table 12 (Figure 4 left) reports the test accuracy of different pruning strategies under weight sparsity levels ranging from 50% to 98%. Across all sparsity regimes, NDC consistently achieves the best performance. While MP, WF, CBS, and SSP maintain competitive accuracy when sparsity is below 80%, their performance begins to deteriorate sharply beyond 90%, and collapses almost completely at 98%, where accuracy falls below 60%. MSP exhibits a more stable behavior, sustaining accuracy above 90% even under 98% sparsity, which highlights its advantage over conventional baselines. Nevertheless, NDC demonstrates a clear margin over all competing methods: at moderate sparsity levels, NDC achieves around 97.8% accuracy, exceeding MSP by 2–3 percentage points and significantly outperforming the other baselines. More strikingly, at extreme sparsity, NDC retains 95.46% accuracy, while MSP drops to 90.73% and the others degrade drastically. These results underline the robustness of NDC in preserving essential representational capacity even under highly compressed regimes, confirming that its Darwinian selection mechanism provides a principled way to identify and preserve critical neurons that remain functionally indispensable after aggressive pruning.

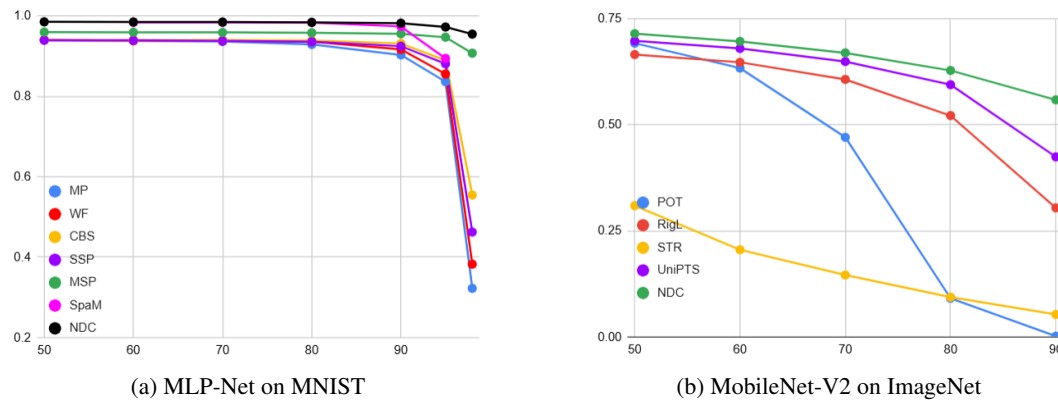

(a) MLP-Net on MNIST        (b) MobileNet-V2 on ImageNet

Figure 4: NDC also outperforms other approaches for MLP-Net on MNIST and MobileNet-V2 on ImageNet. **Left**: MLP-Net on MNIST. **Right**: MobileNet-V2 on ImageNet.

### A.3.6 EXPERIMENT ON CIFAR-10

Figure 5 report the performance of ResNet-18 on CIFAR-10 under different neural and weight sparsity levels, while Figure 6 present analogous results for VGG-16. Across all settings, our proposed NDC consistently outperforms existing approaches. For ResNet-18, NDC achieves superior accuracy at both moderate and extreme sparsity levels. At 90% neural sparsity (Figure 5 left), NDC retains 93.76% accuracy, surpassing the next best method by more than 8%. Similarly, at 98% weight sparsity (Figure 5 right), NDC maintains 89.79%, whereas competing methods degrade below 89%. This demonstrates that NDC preserves critical representational capacity even under aggressive pruning. For VGG-16, NDC again shows consistent advantages. At 90% neural sparsity (Figure 6 left), NDC

achieves 93.27%, whereas the best method falls below 91%. The advantage is even more pronounced in the weight sparsity regime (Figure 6 right), where at 98% sparsity, NDC sustains 92.41% accuracy, compared to drastic performance collapse in EarlySNAP and SNAP. Overall, these results confirm that NDC provides both higher accuracy and greater robustness across architectures and sparsity regimes. The improvements are particularly significant at high sparsity levels, where existing methods fail to retain performance. This indicates that the evolutionary principle underlying NDC—evaluating neurons through NDE, AGC, and NAS—enables the network to preserve essential functional diversity, thereby maintaining generalization despite extreme compression.

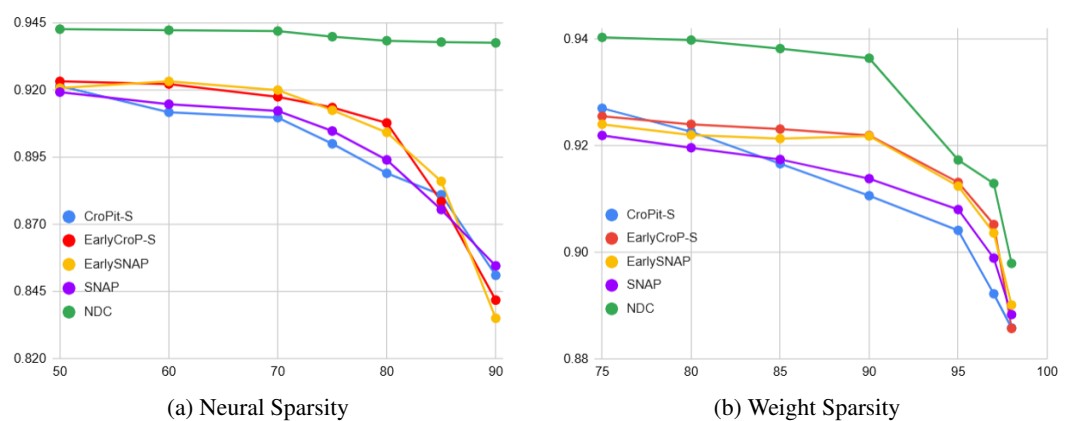

|              (a) Neural Sparsity              |              (b) Weight Sparsity              |

Figure 5: For ResNet-18 networks on CIFAR-10, NDC can find sparser solutions maintaining better performance than other approaches. **Left**: Neural sparsity. **Right**: Weight sparsity.

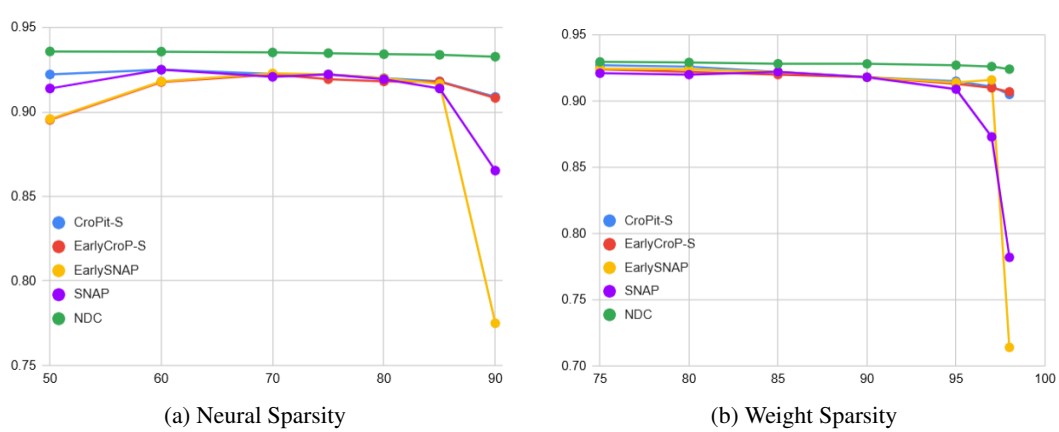

|              (a) Neural Sparsity              |              (b) Weight Sparsity              |

Figure 6: For VGG-16 networks on CIFAR-10, NDC can find sparser solutions maintaining better performance than other approaches. **Left**: Neural sparsity. **Right**: Weight sparsity.

Besides, we evaluate the proposed NDC on the CIFAR-10 dataset using several variants of the ResNet architecture, with the results summarized in Table 13 (Figure 7 and Figure 8 left). Compared with representative pruning approaches, including SNIP Lee (2019), SM, DSR Mostafa & Wang (2019), and DPF Tao (2020), NDC consistently achieves superior performance across a broad spectrum of sparsity levels. Notably, while existing methods exhibit substantial degradation when the sparsity ratio exceeds 90%, NDC demonstrates remarkable robustness, retaining competitive or even enhanced generalization in extremely sparse regimes. For example, on ResNet-20 at 95% sparsity, NDC attains 91.86% accuracy, substantially surpassing DPF and SNIP, while on ResNet-32 and ResNet-56 under the same sparsity level, NDC achieves 91.83% and 93.56%, respectively, both outperforming all competing methods. These gains become increasingly pronounced as the model depth grows, indicating that NDC effectively captures and preserves critical neuron pathways that are indispensable for stable optimization in deeper networks. The empirical results highlight the dual advantages of NDC: its capacity to withstand aggressive pruning without compromising accuracy, and its scalability

Table 13: Top-1 test accuracy on CIFAR-10 for weight pruning.

| Model | Methods | | | | | | Sparsity |
|---|---|---|---|---|---|---|---|
| | SNIP (Lee (2019)) | SM | DSR(Mostafa & Wang (2019)) | DPF(Tao (2020)) | Global MP (Gupta (2024)) | **NDC** | |
| ResNet-20 | 91.32 | 91.99 | 92.19 | 92.56 | * | **92.87** | 70% |
| | 90.80 | 91.70 | 92.06 | 92.38 | * | **92.69** | 80% |
| | 88.63 | 90.16 | 87.92 | 90.95 | * | **92.53** | 90% |
| | 85.16 | 83.77 | * | 88.31 | * | **91.86** | 95% |
| ResNet-32 | 90.66 | 91.72 | 91.64 | 92.60 | 92.70 | **93.47** | 90% |
| | 87.52 | 88.90 | 84.44 | 91.29 | 90.78 | **91.83** | 95% |
| ResNet-56 | 91.77 | 92.94 | 93.98 | 94.06 | * | **94.45** | 90% |
| | * | 91.36 | 92.66 | 92.82 | * | **93.56** | 95% |

to architectures where pruning typically becomes more fragile. Taken as a whole, these observations establish NDC as a robust and adaptable pruning framework that reliably maintains accuracy under stringent resource constraints.

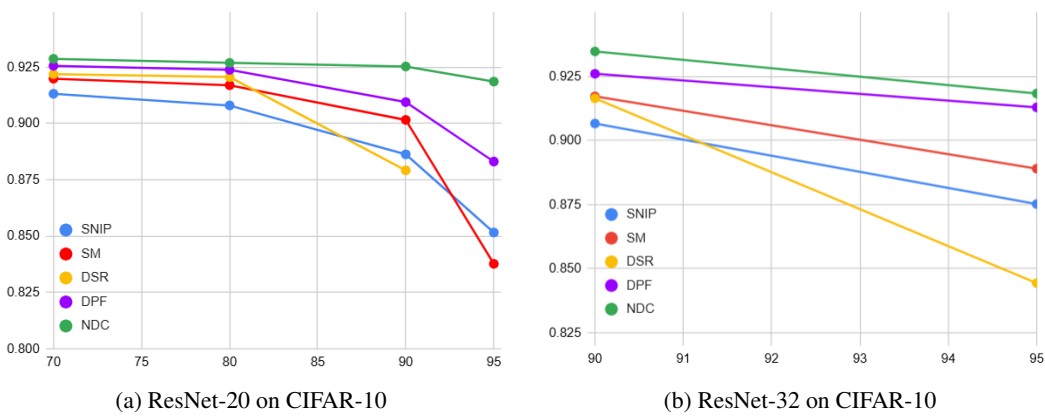

(a) ResNet-20 on CIFAR-10      (b) ResNet-32 on CIFAR-10

Figure 7: NDC also outperforms other approaches for ResNet-20 and ResNet-32 networks trained on CIFAR-10. **Left**: ResNet-20 on CIFAR-10. **Right**: ResNet-32 on CIFAR-10.

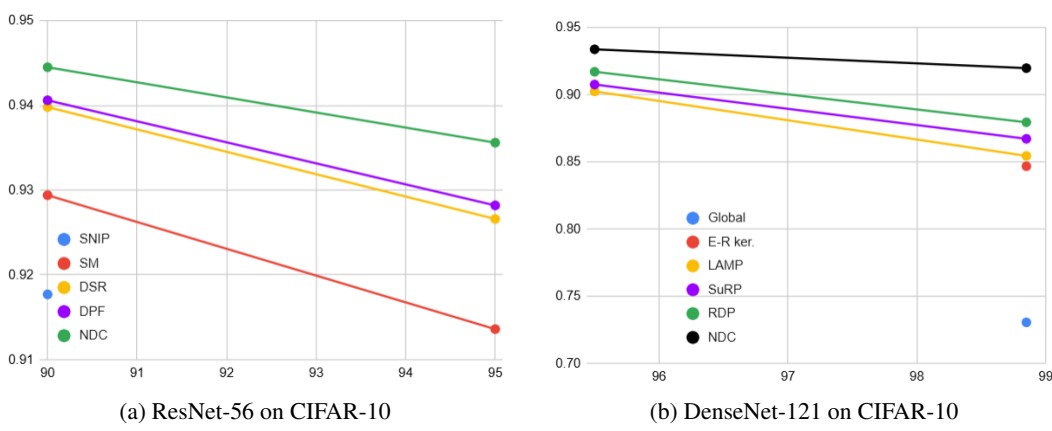

(a) ResNet-56 on CIFAR-10      (b) DenseNet-121 on CIFAR-10

Figure 8: NDC also outperforms other approaches for ResNet-56 and DenseNet-121 networks trained on CIFAR-10. **Left**: ResNet-56 on CIFAR-10. **Right**: DenseNet-121 on CIFAR-10.

### A.3.7 EXPERIMENT ON DENSENET-121

We further evaluate NDC on DenseNet-121 trained on CIFAR-10 under different levels of weight sparsity. As shown in Table 14 (Figure 8 right), NDC consistently achieves superior performance

Table 14: For DenseNet-121 on CIFAR-10. NDC again outperforms the other pruning approachs. **Weight sparsity (%)**.

|  | 95.5 | 98.85 |
|---|---|---|
| Global(Morcos (2019)) | * | $45.30 \pm 27.75$ |
| E-R ker.(Evci (2020)) | * | $59.06 \pm 25.61$ |
| LAMP(Jaeho (2021)) | $90.11 \pm 0.13$ | $85.13 \pm 0.31$ |
| SuRP(Isik (2022)) | 90.75 | 86.71 |
| RDP(Xu (2023)) | $91.49 \pm 0.21$ | $87.70 \pm 0.24$ |
| **Our NDC** | $\mathbf{93.34 \pm 0.03}$ | $\mathbf{91.74 \pm 0.23}$ |

compared to state-of-the-art pruning approaches. At 95.5% sparsity, our method attains 93.34% accuracy, outperforming LAMP, SuRP, and RDP. When the sparsity level is further increased to 98.85%, NDC still preserves a remarkable accuracy of 91.74%, substantially higher than LAMP, SuRP, and RDP. In contrast, earlier unstructured global pruning techniques Morcos (2019); Evci (2020) exhibit unstable performance with large variance, highlighting their lack of robustness under high sparsity constraints. These results demonstrate that NDC is particularly effective at retaining model performance even under extreme compression ratios, thereby confirming its advantage in preserving the critical neurons identified by our selection mechanism.

### A.3.8 EXPERIMENT ON TINY-IMAGENET

We evaluate the effectiveness of our proposed method on ResNet-18 trained on Tiny-ImageNet under varying sparsity levels. Figure 9 compares our approach against established methods. Across all sparsity regimes, NDC achieves consistently higher accuracy, reaching 60.22% at 68.38% sparsity and maintaining 44.46% even at 99% sparsity, representing a clear improvement over the strongest method. This demonstrates the ability of NDC to preserve task-critical neurons under extreme compression. Moreover, NDC attains SOTA efficiency in terms of FLOPs, reducing computation to $0.37 \times 10^8$ at 99% sparsity while simultaneously sustaining higher accuracy than competing methods. Taken together, these results highlight that our method provides a superior trade-off between accuracy and efficiency, outperforming both gradient-based and data-agnostic criteria across all evaluated sparsity levels.

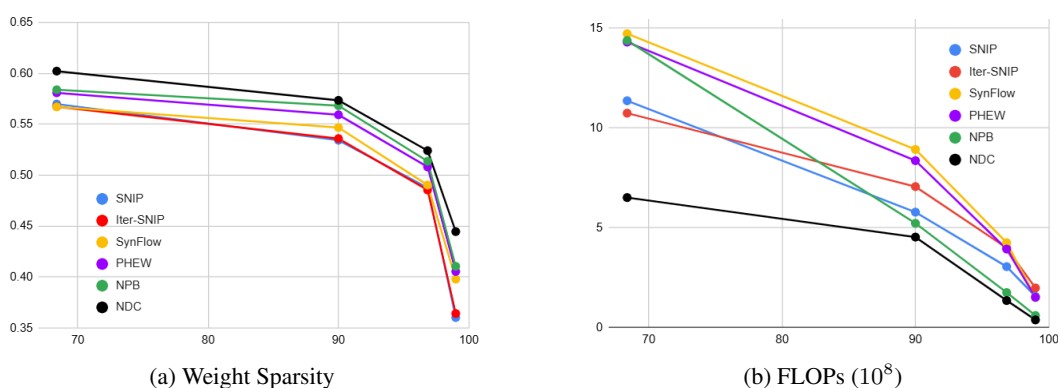

(a) Weight Sparsity          (b) FLOPs ($10^8$)

Figure 9: NDC also outperforms other approaches for ResNet-18 networks trained on Tiny-ImageNet. **Left**: Weight sparsity. **Right**: FLOPs($10^8$).

### A.3.9 EXPERIMENT ON IMAGENET

We further evaluate NDC on the large-scale ImageNet benchmark using MobileNet-V2, a widely adopted architecture for resource-constrained deployment. Figure 4 right presents a comparison against several state-of-the-art pruning frameworks, including POT Lazarevich (2021), RigL Evci (2020), STR Kusupati (2020), and UniPTS Xie (2024), across sparsity levels ranging from 50% to 90%. NDC consistently delivers the best Top-1 accuracy across all sparsity regimes, highlighting its robustness to extreme compression. At moderate sparsity (50%), NDC achieves 71.50%, surpassing UniPTS and RigL. Even under aggressive sparsity (90%), NDC retains 55.90% accuracy, outperform-

ing the next best baseline UniPTS by more than 13 percentage points. These results demonstrate that NDC not only preserves competitive accuracy at low sparsity, but also maintains graceful degradation under severe parameter removal, aligning with the principle of neuronal survival under Darwinian selection pressure. This consistent superiority across compression levels confirms that NDC cultivates representationally diverse and resilient subnetworks, thereby providing a biologically inspired yet practically effective pruning strategy.

### A.3.10 EXPERIMENT ON ILSVRC2012

We evaluate NDC on large-scale ILSVRC2012 image classification using VGG-16 and ResNet-50, and compare it against several state-of-the-art pruning approaches, including ThiNet, CNNpack, and the evolutionary-search–based ECS. Tables 15 and 16 show that NDC consistently achieves the lowest Top-1 and Top-5 error among all baselines under comparable compression budgets. On VGG-16, NDC reduces the Top-1 error to 28.7%, outperforming ECS by 1.1%, while also improving Top-5 accuracy by 0.6% (Table 15). Similar trends hold on ResNet-50, where NDC achieves a Top-1 error of 24.9%, surpassing ECS by 0.9%, and attains the best Top-5 performance (8.0%) among all competing methods (Table 16). These results demonstrate that NDC not only provides competitive compression ratios but also preserves accuracy more effectively across architectures with different connectivity patterns.

Regarding the reviewer's comment on the KDD 2018 work "Towards Evolutionary Compression", we note that both their method and ours employ evolutionary principles to guide network pruning. However, the two approaches differ fundamentally in search granularity and optimization objectives. Evolutionary Compression performs population-based search directly over structured pruning config- urations, treating each candidate as an independent architecture. In contrast, NDC introduces a neural dependency–aware compression mechanism that embeds layer-wise dependency structures into the evolutionary process, enabling more informed mutation and selection. This design substantially reduces the search space and alleviates instability during evolution, which we believe contributes to the performance gains observed in Tables 15 and 16. We have now included this discussion in the revised version for clarity.

Table 15: VGG-16 on ILSVRC2012 — Top-1 and Top-5 error rates (%) for SOTA pruning methods, including ThiNet Luo (2017), CNNpack Wang (2016) and ECS Wang (2018).

|  | Methods | | | |
| --- | --- | --- | --- | --- |
|  | ThiNet | CNNpack | ECS | NDC (Ours) |
| Top-1 Err (%) | 30.2 | 30.5 | 29.8 | **28.7** |
| Top-5 Err (%) | 11.0 | 11.2 | 10.7 | **10.1** |

Table 16: ResNet-50 on ILSVRC2012 — Top-1 and Top-5 error rates (%) for SOTA pruning methods.

|  | Methods | | | |
| --- | --- | --- | --- | --- |
|  | ThiNet | CNNpack | ECS | NDC (Ours) |
| Top-1 Err (%) | 26.0 | 26.4 | 25.8 | **24.9** |
| Top-5 Err (%) | 8.9 | 9.1 | 8.4 | **8.0** |

### A.3.11 LEAKY RELU

Our pruning framework was initially designed with *ReLU* activations, where neurons can become permanently inactive due to the zero-gradient regime. Such inactive neurons naturally align with the principle of Darwinian elimination, providing clear pruning candidates. By contrast, *Leaky ReLU* Kou (2023); Karhadkar (2024) introduces a small negative slope, ensuring that neurons never completely die. This property complicates pruning, as all neurons remain technically active even if their contributions are marginal. To examine whether NDC can adapt to this setting, we hypothesize that neurons whose activations lie almost entirely within the negative regime across representative

Table 17: ResNet-18 networks with *Leaky ReLU* trained on CIFAR-10. NDC again outperforms the other pruning methods. **Neural sparsity (%)**.

|  | 50 | 60 | 70 | 75 | 80 | 85 | 90 |
|---|---|---|---|---|---|---|---|
| EarlyCroP-S | 88.89 | 88.97 | 88.17 | 87.10 | 85.99 | 84.72 | 80.71 |
| EarlySNAP | 89.40 | 87.86 | 87.02 | 85.99 | 85.00 | 84.33 | 80.00 |
| SNAP | 88.21 | 87.62 | 86.67 | 86.07 | 85.48 | 81.75 | 78.33 |
| **NDC** | **94.40** | **94.39** | **94.28** | **94.13** | **94.01** | **93.95** | **93.94** |

Table 18: ResNet-18 networks with *Leaky ReLU* trained on CIFAR-10. NDC again outperforms the other pruning methods. **Weight sparsity (%)**.

|  | 75 | 80 | 85 | 90 | 95 | 97 | 98 |
|---|---|---|---|---|---|---|---|
| EarlyCroP-S | 88.95 | 88.90 | 88.94 | 88.78 | 87.56 | 86.34 | 85.48 |
| EarlySNAP | 89.33 | 89.39 | 88.78 | 87.72 | 86.66 | 85.59 | 84.78 |
| SNAP | 88.34 | 88.22 | 87.85 | 87.29 | 86.27 | 85.39 | 83.32 |
| **NDC** | **94.25** | **94.17** | **93.91** | **93.66** | **90.31** | **90.17** | **90.07** |

minibatches provide negligible contributions to the model's predictive capacity. We therefore extend NDC to identify and prune such weakly contributing neurons. Empirical results on ResNet-18 trained with *Leaky ReLU* on CIFAR-10 are summarized in Tables 17 and 18 (Figure 10). Under both neural sparsity and weight sparsity regimes, NDC consistently and significantly outperforms competitive methods. For instance, at 90% neural sparsity, NDC retains an accuracy of 93.94%, compared to only 80.71% for EarlyCroP-S and 80.00% for EarlySNAP. Similarly, at 95% weight sparsity, NDC achieves 90.31% accuracy, outperforming EarlySNAP and SNAP. These results demonstrate that NDC generalizes beyond hard-threshold activations such as ReLU, effectively exploits functional inactivity in Leaky ReLU neurons, and consistently yields superior sparsity–accuracy trade-offs, confirming the robustness of Darwinian selection principles across different activation functions.

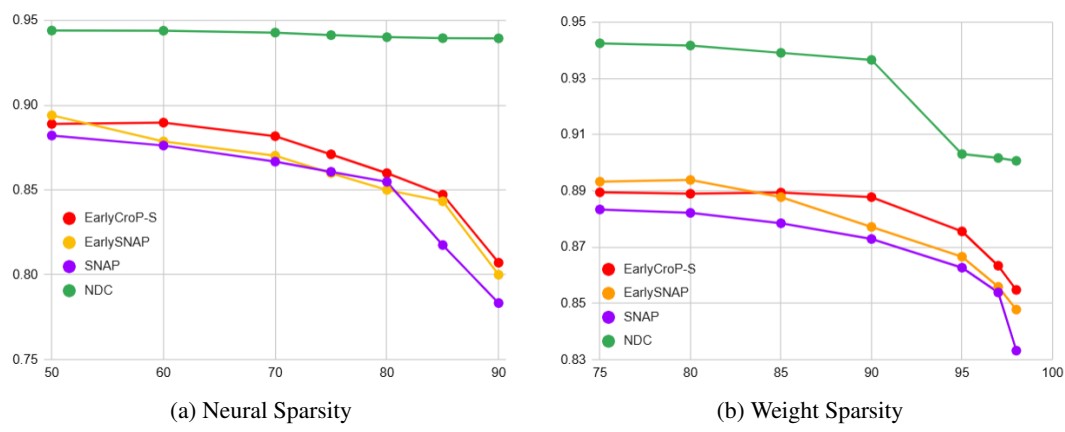

(a) Neural Sparsity  (b) Weight Sparsity

Figure 10: ResNet-18 networks with *Leaky ReLU* trained on CIFAR-10. NDC again outperforms the other pruning methods. **Left**: Neural sparsity. **Right**: Weight sparsity.

