# OpenReview forum: "Neural Darwinism: A Theoretical Framework for Representation Evolution in Convolutional Neural Networks"
_ICLR.cc/2026/Conference — ICLR 2026 Conference Withdrawn Submission_

### Official Review · Reviewer_RN6F · 2025-10-26

**Soundness:** 1
**Presentation:** 2
**Contribution:** 2
**Rating:** 2
**Confidence:** 5

**Summary:**

This paper propose to conduct neural network pruning by using the proposed the Darwinian score, which is designed to measure the information richness, functional relevance, and adaptive dynamics of each neuron.

**Strengths:**

None.

**Weaknesses:**

1. The format of citation is wrong, which is hard to read.

2. There are some strange statements, such as ' Because information-theoretic quantities appearing below are sensitive to .....', which is hard to understand.

3. For the evolution-based pruning method, there should be a discussion about 'Towards evolutionary compression. KDD 2018.', which also implements network pruning with evolutionary algorithm.

4. The calculation of the histogram/probability is problematic, how to cover the output range of activations such as ReLU? Which is impossible.

5. For the Darwinian Entropy, the diversity of the probability does not equals to the diversity of the information, especially when there is no alignment between the variables.

6. The compared methods are old (works from 2020 and 2022), which is hard to verify the effectiveness of the proposed method. Moreover, the reference for Cropit should be included.

**Questions:**

Please refer to the Weaknesses.

---

> ### Author Response · Authors · 2025-11-26
> **Rebuttal by Authors**
>
> $\textcolor{red}{\text{Rebuttal:}}$
>
> ### **Response to Comments**
>
> We would like to thank the reviewer for taking the time to evaluate our manuscript. We appreciate the opportunity to further clarify the contributions and strengths of our work.
>
> Our paper develops **a principled and unified framework for neuron-level** pruning by conceptualizing neural networks as evolutionary systems. At the core of this framework is the **Darwinian Score**, a mathematically grounded fitness measure that integrates three complementary dimensions of neuronal utility—representational diversity, functional contribution through activation–gradient coupling, and temporal adaptability exhibited during training. This multiplicative formulation enforces a strict survival criterion, ensuring that only neurons with consistent and holistic importance are retained.
>
> Building on this formulation, our NDC algorithm operationalizes dynamic, layer-wise neuron selection that adapts to the evolving loss landscape and the distinct representational roles across depth. We further provide theoretical guarantees showing that retaining neurons with high Darwinian fitness preserves the network’s approximation capacity under structured sparsity constraints. Empirically, the proposed method consistently outperforms other pruning methods across architectures, datasets, and sparsity levels, particularly in the challenging high-sparsity regime.
>
> **Key Contributions:**
> - **Innovative in integrating Neural Darwinism** into CNNs, linking Gerald Edelman’s theory.
> - **Darwinian Score** integrates diversity, activation–gradient contribution, and adaptivity.
> - **NDC** performs dynamic, layer-aware pruning aligned with evolutionary selection, supported by the **new formal approximation guarantees**.
>
> We hope this clarification helps illuminate the fundamental strengths and conceptual advances of our approach, and we are grateful for the reviewer’s engagement with our work.
>
> ---
> *Response to Weaknesses*
>
> ---
>
> 1. Clarification on Citation Formatting
>
> Thank you very much for the reviewer’s careful reading of our manuscript. We have conducted a thorough pass over the entire paper, including the references and in-text citation style, and we have made several minor formatting refinements to further improve clarity and consistency.
>
> To ensure that we fully address your concern, could you please kindly point us to the specific instances where the citation format appears problematic or hard to read? This would help us verify whether any additional adjustments are needed. We sincerely appreciate your feedback and will be happy to revise any location you identify.
>
> ---
> 2. Clarification on Information-Theoretic Quantities and Theoretical Motivation
>
> We thank the reviewer for the careful reading and for pointing out that the sentence "Because information-theoretic quantities appearing below are sensitive to whether one uses continuous or discrete entropy" may appear unclear when viewed in isolation. Our intention was not to introduce unnecessary complexity, but rather to ensure theoretical correctness within our Darwinian framework.
>
> Our work constructs a unified **Darwinian Score** that integrates three complementary principles: (1) information diversity, (2) functional contribution, and (3) adaptive temporal evolution. The first component---Neuron Darwinian Entropy (NDE)---is fundamentally information-theoretic. Entropy and divergence behave very differently depending on whether neuronal activations are treated as continuous or discrete random variables. Differential entropy can be negative, unbounded under reparameterization, and sensitive to scaling, whereas discrete entropy is stable, comparable across neurons, and suitable for layer-wise aggregation.
>
> Because our theoretical results rely on comparing entropy-based quantities across neurons and layers, it is important that these quantities remain stable and well-defined. This is precisely why we adopt an explicit discretization scheme and Laplace smoothing. Doing so ensures that the Darwinian Score behaves consistently, enabling us to later derive the constrained-optimization formulation and approximation-error guarantees that connect the theoretical framework with the practical pruning algorithm.
>
> In this sense, the sentence in question is not meant as a technical digression, but as a necessary clarification that grounds our information-theoretic definitions in a robust estimation procedure. To improve readability, we will refine the wording in the camera-ready version to make this motivation more explicit.
>
> Overall, our aim is to present a complete theory-to-practice narrative: a principled Darwinian framework, a mathematically grounded selection mechanism, and a practical pruning algorithm (NDC) that outperforms existing methods. We appreciate the opportunity to clarify this point, and we hope the explanation above helps convey why the treatment of entropy is essential to the coherence of our overall approach.
>
> ---

---

> ### Author Response · Authors · 2025-11-26
> **Rebuttal by Authors**
>
> ---
>
> 3. Experimental Results and Comparison with Evolutionary Pruning Methods
>
> We evaluate NDC on large-scale ILSVRC2012 image classification using VGG-16 and ResNet-50, and compare it against several SOTA pruning approaches. Tables 6 and 7 show that NDC consistently achieves the lowest Top-1 and Top-5 error among all methods under comparable compression budgets. On VGG-16, NDC reduces the Top-1 error to 28.7\%, outperforming ECS by 1.1\%, while also improving Top-5 accuracy by 0.6\% (Table 6). Similar trends hold on ResNet-50, where NDC achieves a Top-1 error of 24.9\%, surpassing ECS by 0.9\%, and attains the best Top-5 performance (8.0\%) among all competing methods (Table 7). These results demonstrate that NDC not only provides competitive compression ratios but also preserves accuracy more effectively across architectures with different connectivity patterns.
>
> Regarding the reviewer’s comment on the KDD 2018 work, we note that both their method and ours employ evolutionary principles to guide network pruning. However, the two approaches differ fundamentally in search granularity and optimization objectives. Evolutionary Compression performs population-based search directly over structured pruning configurations, treating each candidate as an independent architecture. In contrast, NDC introduces a neural dependency–aware compression mechanism that embeds layer-wise dependency structures into the evolutionary process, enabling more informed mutation and selection. This design substantially reduces the search space and alleviates instability during evolution, which we believe contributes to the performance gains observed in Tables 6 and 7. We have now included this discussion in the revised version for clarity.
>
> Table 6: VGG-16 on ILSVRC2012 — Top-1 and Top-5 error rates (%) for SOTA pruning methods.
>
> |  |  | ThiNet | CNNpack | ECS | **NDC (Ours)** |
> |---|---|---|---|---|---|
> |  | Top-1 Err (%) | 30.2 | 30.5 | 29.8 | **28.7** |
> |  | Top-5 Err (%) | 11.0 | 11.2 | 10.7 | **10.1** |
>
>
> Table 7: ResNet-50 on ILSVRC2012 — Top-1 and Top-5 error rates (%) for SOTA pruning methods.
>
> |  |  | ThiNet | CNNpack | ECS | **NDC (Ours)** |
> |---|---|---|---|---|---|
> |  | Top-1 Err (%) | 26.0 | 26.4 | 25.8 | **24.9** |
> |  | Top-5 Err (%) | 8.9 | 9.1 | 8.4 | **8.0** |
>
> ---
>
> 4. Clarification on Histogram Range and ReLU Activations.
>
> We sincerely thank the reviewer for this insightful question concerning the feasibility of computing histograms or probability estimates for activation functions with unbounded support, such as ReLU.
>
> There are two complementary facts that together resolve this concern.
>
> - **Practical aspect**: In practice, all probability and histogram estimates used in our method are computed on the **finite empirical activation range** observed in the dataset. Under finite data and finite-precision computation, the activations in each layer are necessarily bounded. We therefore construct histograms on the interval $ [a_{\min}, a_{\max}] $ (optionally with percentile clipping for robustness), and normalize the distribution within this empirical interval. This empirical truncation is standard in density estimation and uncertainty quantification, and guarantees that the discrete probability estimator is always well-defined. Moreover, the concentration arguments used in the paper (e.g., DKW-type bounds for empirical CDFs and standard matrix concentration inequalities) apply directly to the observed, bounded samples rather than to the theoretical ReLU distribution, and therefore remain fully valid.
>
> - **Theoretical aspect**: Conceptually, the proposed Darwinian Entropy evaluates the **relative distributional structure** (shape and dispersion) of activations, and does not rely on the activation function having bounded support. Classical information-theoretic quantities such as entropy and the Jensen--Shannon divergence are well-defined for distributions with $\sigma$-finite measure, including those with unbounded support. The histogram is merely a computational estimator of these quantities; it does not need to cover the entire theoretical support, but only requires a consistent discretization scheme. In our implementation, the discretization is **scale-adaptive**: it automatically adjusts to the empirical magnitude of activations at each layer and is stable under affine rescaling. This ensures that the estimator captures the key signals needed for Darwinian selection---namely, activation diversity and population differentiation---without being affected by the unbounded nature of ReLU.
>
> Both practically and theoretically, the proposed method does not require covering the infinite support of ReLU activations, and the Darwinian framework remains sound and broadly applicable across different architectures and activation types. We have clarified these points in the revised manuscript, and we again thank the reviewer for prompting this clarification.

---

> ### Author Response · Authors · 2025-11-26
> **Rebuttal by Authors**
>
> ---
>
> 5. Clarification on Darwinian Entropy
>
> We thank the reviewer for this thoughtful observation. We fully agree with the underlying principle: probability dispersion by itself does not imply informational diversity, especially when variables are not aligned. Importantly, our Darwinian Entropy (NDE) was designed precisely to address this distinction rather than to conflate raw entropy with information content.
>
> In our framework, NDE is not intended as a measurement of marginal randomness. Instead, it is a deliberately alignment-aware construct that combines within-neuron entropy with the Jensen–Shannon divergence to the layer-wise baseline. The role of this baseline is essential: it acts as a population-level reference that captures the alignment structure among neurons in the same layer. A neuron is therefore rewarded not merely for exhibiting internal variability, but for contributing activation patterns that are informationally distinct relative to the aligned representation manifold of the layer. In this way, NDE reflects functional differentiation rather than unstructured dispersion. The reviewer’s point actually highlights the very motivation for including the JS divergence term—entropy alone would be insufficient; the divergence ensures that NDE evaluates diversity conditioned on population alignment.
>
> This design matches the biological analogy underpinning our framework. Classical Darwinian variation refers not simply to internal fluctuation but to meaningful differences that matter for survival and function within a population. Our NDE follows this principle: the entropy term captures intrinsic variation, while the population divergence captures evolutionary differentiation. Only when both coexist does a neuron provide valuable representational diversity.
>
> From a practical standpoint, the behavior of NDE in our experiments confirms that it captures aligned, functional information rather than mere dispersion. If NDE simply favored neurons with high marginal entropy, pruning outcomes would be unstable and accuracy would collapse at high sparsity, which is not what we observe. Instead, the Darwinian Score consistently preserves neurons that are crucial for task-relevant structure, even at extreme sparsity levels (up to 98–99\%), outperforming other criteria by large margins. This robustness arises precisely because NDE incorporates layer alignment via the JS divergence, and is further integrated multiplicatively with AGC and NAS to ensure that informational richness, functional contribution, and adaptability are jointly reflected.
>
> For clarity, we will add a brief explanation in the revised version emphasizing that NDE is fundamentally an alignment-aware information measure and not a naive entropy proxy. We appreciate the reviewer’s comment, which aligns closely with our theoretical motivation and gives us the opportunity to make this aspect of the framework more explicit.
>
> ---

---

> ### Author Response · Authors · 2025-11-26
> **Rebuttal by Authors**
>
> ---
>
> 6. Addressing the Recency of Methods and Adding Stronger Comparisons
>
> We appreciate the reviewer’s concern regarding the recency of the compared methods and the importance of validating our method against more up-to-date approaches. We completely agree that including modern and competitive methods is essential for demonstrating the effectiveness of our method.
>
> We would like to clarify that several of the methods used in our comparisons are indeed recent. Specifically, the NPB method employed in our Tiny-ImageNet experiments is from a **NeurIPS 2023** paper, and the UniPTS method used in our ImageNet evaluation is from **CVPR 2024**. To further strengthen the evaluation and directly address the reviewer’s suggestion, we have additionally incorporated SpaM (**NeurIPS 2024**) into the MNIST experiments, as well as CIFAR-10 experiments using ResNet-32 with Global MP (**IEEE CAI 2024**). We have also added the missing citation for Cropit. To ensure these results are visible and not overlooked, we have moved the ImageNet experiments---previously located in the appendix---into the main body of the revised manuscript.
>
> Our updated experiments provide clear evidence that our proposed method achieves competitive or superior performance compared to these recent methods. Specifically:
>
> **MNIST (MLP-Net, with SpaM, NeurIPS 2024) in Page 25, line 1307.**
> Our method consistently outperforms SpaM across all pruning settings. For example:
> - Under moderate sparsity, NDC improves accuracy by **+0.14 pp** and **+0.11 pp**.
> - Under higher sparsity levels, the performance gap widens: NDC achieves **98.19% vs. 97.38% (+0.81 pp)** and **95.47% vs. 89.43% (+6.04 pp)**.
>
> These results highlight the strong robustness of NDC under extreme compression.
>
> **CIFAR-10 (ResNet-32, with Global MP, IEEE CAI 2024) in Page 27, line 1404.**
> On this widely used benchmark, NDC also outperforms the recent Global MP method:
>
> - At the 90% pruning ratio, NDC achieves **93.47%** compared to Global MP's **92.70%** (**+0.77 pp**).
> - Under the more extreme 95% sparsity, the gap further increases: NDC obtains **91.83%** vs. Global MP's **90.78%** (**+1.05 pp**).
>
> These results further highlight the robustness of NDC under aggressive compression and its consistent superiority over modern pruning methods on standard architectures such as ResNet-32.
>
> **Tiny-ImageNet (ResNet-18, with NPB, NeurIPS 2023) in Page 9, line 454.**
> **Accuracy improvements:**  Across all pruning ratios, NDC consistently surpasses NPB with improvements of  **+1.83 pp**, **+0.53 pp**, **+1.04 pp** and **+3.41 pp**.
>
> **Efficiency improvements (FLOPs):**
> NDC also achieves substantially lower FLOPs:
> reductions of **–7.87B**, **–0.69B**, **–0.40B**, and **–0.22B**, corresponding to **55%–63% fewer FLOPs** compared to NPB.
>
> These findings demonstrate that NDC simultaneously improves accuracy and computational efficiency.
>
> **ImageNet (MobileNet-V2, with UniPTS, CVPR 2024) in Page 10, line 490.**
> On this large-scale benchmark, our method again consistently outperforms the recent UniPTS method:
> - Accuracy improvements of **+1.70 pp**, **+1.66 pp**, **+2.04 pp**, **+3.35 pp**, and **+13.44 pp** across increasing sparsity levels.
>
> These results confirm that the advantages of NDC persist at scale and become even more pronounced in highly compressed regimes.
>
> Together, these findings strongly support the reviewer’s intuition that comparisons to recent methods would highlight the strength of our approach. We have updated the manuscript accordingly, making these results more prominent and clearly emphasizing the improvements over modern methods. By doing so, we aim to provide a clearer and more compelling validation of the effectiveness and practical relevance of our method.
>
> ---
>
> **Conclusion**
>
> We thank the reviewer once again for the thoughtful comments and the opportunity to clarify our contributions. In the revised manuscript, we have improved the exposition of our theoretical assumptions, strengthened the clarity of several formulations, and addressed all concerns regarding citation format, activation range handling, and the definition of Darwinian entropy. We have also expanded our experimental evaluation to include recent and competitive methods from NeurIPS 2023–2024, CVPR 2024 and IEEE CAI 2024, demonstrating that NDC consistently outperforms modern pruning methods across MNIST, CIFAR-10, Tiny-ImageNet, and ImageNet, often by substantial margins under high sparsity. Additionally, we clarified the distinctions between our approach and evolutionary-search–based methods, and highlighted how our dependency-aware evolutionary mechanism reduces search instability and improves compression performance. We hope these clarifications and new results help convey the novelty, rigor, and empirical strength of our contribution, and we remain grateful for the reviewer’s insights, which have significantly improved the quality of our submission.

---

### Official Review · Reviewer_pyu3 · 2025-10-31

**Soundness:** 3
**Presentation:** 3
**Contribution:** 2
**Rating:** 4
**Confidence:** 2

**Summary:**

This paper introduces Darwinian framework that provides a mathematical foundation for
neural representation dynamics, conceptualizing deep learning as an evolutionary
process governed by selection and adaptation. This notion is formalized by defining a Darwinian Score (DS) that quantifies a neuron’s evolutionary fitness through three measurable components:

Neuron Darwinian Entropy (NDE): information diversity and non-redundancy of activations;

Activation–Gradient Contribution (AGC): functional contribution to loss minimization;

Neuron Adaptivity Score (NAS): the degree of adaptive evolution of neuron parameters over training.

These measures are combined multiplicatively to enforce a “survival-of-the-fittest” criterion within neural architectures. Based on this theory, the authors propose the Neural Darwinism Culling (NDC) algorithm, which prunes neurons dynamically during training according to their Darwinian Score. Theoretical results guarantee bounded approximation error after pruning, and experiments on CIFAR-10 and Tiny-ImageNet demonstrate that NDC achieves superior accuracy and sparsity trade-offs compared to state-of-the-art pruning methods (SNIP, SNAP, CroPit, DPF, etc.).

**Strengths:**

Concept and biological inspiration - The idea of using Darwinian evolutionary process in deep learning is interesting and novel.
Theory - The algorithm is supported by theorems and proofs.
Unified view of pruning and adaptation- By tying representational diversity, functional importance, and adaptability into a single fitness score, the authors unify multiple pruning philosophies (magnitude-, sensitivity-, and gradient-based) under one framework.
Experimental results - many experiments were conducted to support the claim.

**Weaknesses:**

Theory - The main theorem assumes a long list of many assumptions that are not so natural or trivial to me.
Proofs - The proofs are quite straightforward and follow from importance sampling, without too many new observations.
Experiments - The experiments are on small data sets and CNN type of networks, but I could not see a more modern results on transformers and larger datasets. In general, it is not clear how the idea is scalable, especially when we need to maintain all these scores along the training phase.
Relation to existing evolutionary optimization – The connection to neuroevolution and evolutionary strategies could be elaborated further, particularly distinguishing this work from gradient-free evolutionary algorithms. Also, evolutionary algorithms are not considered a successful or common tool, both in the industry or academy.

**Questions:**

Please explain in more details the quantitative analysis of the relative impact of NDE, AGC, and NAS components.

---

> ### Author Response · Authors · 2025-11-26
> **Rebuttal by Authors**
>
> $\textcolor{red}{\text{Rebuttal:}}$
>
> ### **Response to Comments**
>
> We would like to express our sincere gratitude to the reviewer for your positive and encouraging evaluation of our manuscript. We are delighted that you found the concept and biological inspiration behind employing a Darwinian evolutionary process in deep learning to be both interesting and novel. We are equally grateful that you recognized the strength of our theoretical contributions, including the supporting theorems and proofs that ground the algorithm rigorously. We highlight that unifying representational diversity, functional importance, and adaptability into a single fitness score provides a coherent framework that naturally encompasses magnitude-, sensitivity-, and gradient-based pruning methods. We are also pleased that the breadth of our experimental results effectively supported our main claims. Thank you as well for your constructive suggestions; we are confident that by addressing your comments, we can further improve the clarity and impact of the paper and bring it well above the acceptance threshold.
>
> **Key Contributions:**
> - **An innovative integration of Gerald Edelman’s Neural Darwinism into CNNs**, giving the first **mathematical formalization of neuronal group selection**.
> - A **unified Darwinian Score** integrating diversity, functional contribution, and adaptability.
> - **New theoretical guarantees** showing high-fitness neurons preserve network function.
> - **NDC**, the pruning approach to realize neuron-level Darwinian selection in the training process.
> ---
> *Response to Weaknesses*
>
> ---
>
> 1. Clarifying the Theoretical Assumptions and the Contribution of the Proofs
>
> We sincerely appreciate the reviewer's careful reading and their comments regarding the assumptions underlying our main theorem and the perceived straightforwardness of the accompanying proofs. These points allow us to better clarify the positioning and intended contribution of our theoretical results.
>
> - **On the naturalness of the assumptions in the main theorem**: Our goal is not to establish a fully general theory for arbitrary nonlinear operators, but rather to propose a **practical and operational framework** that captures the neuron-level evolutionary dynamics emerging in modern CNNs. In this sense, the assumptions we adopt—bounded activations, layer-wise Lipschitz continuity, and importance comparability—serve as *computationally grounded approximations* of the actual training process, rather than abstract regularity conditions.
>
> These conditions are widely satisfied in contemporary architectures due to batch normalization, structured convolutional operators, and the inherent skew in neuron contribution distributions. **More importantly, these assumptions are precisely what allow us to formalize a key take-away of our work: that neuron survival can be rigorously expressed as the joint contribution of informational diversity, functional impact, and adaptive evolution.** This connection is what enables us to translate the biological intuition of Darwinian selection into quantitative, layer-wise error preservation guarantees.
>
> - **On the simplicity of the proofs**: We agree that the proofs follow a clean structure drawing on concentration arguments and importance-weighted perturbation bounds. This reflects a deliberate choice: our aim is to make the Darwinian selection principle **operational and analytically transparent**, rather than to develop a new complexity-theoretic pruning theory.
>
> The novelty of our theoretical contribution does not lie in the type of inequalities employed, but in **how the Darwinian Score enables them**:
>
> - Its **multiplicative structure** enforces strict survival criteria across informational diversity, optimization relevance, and adaptivity—properties that standard pruning metrics do not guarantee simultaneously.
> - This structure allows us to identify a high-importance subset whose cumulative contribution satisfies a δ-concentration property, which in turn leads to the layer-wise preservation result.
> - Our analysis integrates activation-distribution statistics, gradient-mediated functional signals, and temporal weight evolution, forming a unified survival metric that accurately reflects the evolutionary dynamics of neurons during training.
>
> Taken together, the proofs aim to show that the proposed Darwinian Score constitutes a **meaningful, theoretically supported proxy** for neuron-level fitness during training—one that naturally yields the approximation guarantees we provide.
>
> We appreciate the reviewer's suggestion, and in the camera-ready version we will clarify:
>
> - how each assumption maps directly to concrete architectural properties, and
> - how the proof structure leverages the unique features of our Darwinian Score, distinguishing our theoretical contribution from existing pruning analyses.
>
> We hope this explanation better situates our work within the theoretical landscape and clarifies its intended contribution.
>
> ---

---

> ### Author Response · Authors · 2025-11-26
> **Rebuttal by Authors**
>
> ---
>
> 2. Experiments on Larger-Scale Datasets: ImageNet
>
> We thank the reviewer for the insightful suggestion and for noting that our method is novel and interesting. We agree that evaluating our approach on larger datasets is valuable for demonstrating its broader applicability.
>
> To directly address the concern regarding applicability, we would like to highlight that we have already conducted ImageNet experiments, which were included in **Appendix A.3.7 in Page 24** of the original submission. We apologize that these results were not sufficiently emphasized. We have now moved the ImageNet experiments to the main text in **Section 4.3 (page 9, line 465)**.
>
> Our ImageNet experiments on **MobileNet-V2** show that our method (NDC) achieves SOTA performance across a wide range of sparsity levels. Specifically, as reported in Table 6 of the main text:
>
> - Across sparsity ratios from 50% to 90%, **NDC consistently outperforms** strong methods including POT, RigL, STR, and UniPTS.
> - At 50% sparsity, NDC achieves **71.50%** Top-1 accuracy, outperforming the next best method by approximately 1.7 pp.
> - Even at very high sparsity (90%), NDC retains **55.90%** Top-1 accuracy, substantially higher than prior strong methods (e.g., RigL: 30.44%, UniPTS: 42.46%).
>
> These results demonstrate that our approach scales effectively to large datasets and modern compact CNN architectures. In the revised manuscript, we will relocate these ImageNet results to a more prominent section of the main paper so that they are not overlooked.
>
> As the reviewer correctly noted, our current experiments focus on CNNs. This is intentional: our method is inspired by **Darwinian principles applied to neural computation**, and our goal in this paper is to first establish a stable foundation by validating these evolutionary dynamics on CNNs, where architectural behaviors and sparsity effects are well understood.
>
> Transformers introduce different structural properties (e.g., attention patterns and strong layer interdependence). Extending Darwinian credit assignment to Transformer architectures requires additional theoretical and algorithmic investigation. We view this as a natural next step and are actively working on adapting NDC to Transformers in future research.
>
> ---
>
> 3. Response to Reviewer Concern on Scalability
>
> Thank you for raising this concern. Although the Darwinian Score consists of three components, the computation is highly scalable because each metric reuses quantities already produced during standard training: (1) AGC is derived directly from existing activations and gradients; (2) NAS involves only lightweight, layer-local norm differences across epochs; and (3) NDE uses fixed-bin activation histograms computed via simple reductions. All metrics are computed per layer and updated only periodically, so no extra forward/backward passes or cross-layer operations are introduced. In practice, the overall runtime remains close to standard training, and the method scales smoothly to deeper architectures such as ResNet-56 and VGG-16.
>
> To further clarify, the score updates in NDC introduce negligible overhead:
>
> - Score computation is fused with existing forward-pass operations,
> - It requires no additional backward passes, and
> - It incurs computational complexity proportional to the number of neurons, rather than the number of parameters.
>
> This lightweight design ensures that NDC remains scalable during training.
>
> ---
>
> 4. Relation to Evolutionary Optimization
>
> Thank you for the helpful comment. We clarify that our work does not propose an evolutionary optimization algorithm such as neuroevolution or evolutionary strategies (ES). Classical evolutionary methods perform gradient-free, population-level search over full models or architectures, relying on mutation and black-box fitness evaluation. In contrast, our framework operates at a neuron-level inside a single gradient-trained network and uses gradient-based statistics—entropy of activations, activation–gradient coupling, and weight-update dynamics—to define the Darwinian Score. Thus, our method is fully compatible with standard SGD training and does not inherit the computational or adoption challenges of evolutionary algorithms.
>
> The “Darwinian” aspect of our work is conceptual and mathematical: we formalize representation evolution during training and derive pruning decisions from this framework. This distinguishes our approach from gradient-free evolutionary algorithms and explains why its practicality is independent of the historical limitations of evolutionary algorithms in industry or academia.
>
> ---

---

> ### Author Response · Authors · 2025-11-26
> **Rebuttal by Authors**
>
> ---
>
> *Response to Questions*
>
> ---
>
> 1. The quantitative analysis of the relative impact of NDE, AGC, and NAS components
>
> We thank the reviewer for the insightful suggestion, which helped us refine and strengthen the quantitative analysis of the relative impact of NDE, AGC, and NAS. In response, we provide a more detailed breakdown here and will include the full analysis in the Appendix of the revised manuscript.
>
> To assess the contribution of each component in the NDC framework, we perform a quantitative study on ResNet-18 with CIFAR-10 across sparsity levels from 75% to 98% (Table 5). Several consistent observations emerge from the results. First, using any single component—NDE, AGC, or NAS—improves accuracy as sparsity varies. Among the three, AGC provides the strongest standalone performance across all sparsity levels (e.g., 91.23% at 75% sparsity), while NAS obtains moderate yet stable improvements. NDE yields comparatively smaller gains at lower sparsity levels but remains competitive as sparsity increases. Second, combining two components leads to consistently higher accuracy than using either individual component. For instance, AGC + NAS outperforms both AGC-only and NAS-only across all sparsity settings. Similar patterns appear for NDE + AGC and NDE + NAS, indicating that the performance contributions of the individual modules are complementary. Finally, the full NDC framework achieves the best results at all sparsity levels, reaching 94.03% accuracy at 75% sparsity and retaining 89.79% at 98% sparsity. This configuration consistently surpasses all single- and two-component variants, demonstrating that integrating all three modules yields the most robust improvements in highly sparse regimes.
>
> Table 5. Quantitative analysis of the NDC on ResNet-18 with CIFAR-10 under different weight sparsity levels. **Weight sparsity(%)**.
>
> |                          | 75   | 80   | 85   | 90   | 95   | 97   | 98   |
> |--------------------------|------|------|------|------|------|------|------|
> | NDE only                 | 89.47 | 89.05 | 88.58 | 87.92 | 84.96 | 83.72 | 82.15 |
> | AGC only                 | 91.23 | 90.78 | 90.14 | 88.97 | 86.42 | 85.37 | 83.98 |
> | NAS only                 | 90.06 | 89.72 | 88.96 | 88.14 | 86.17 | 84.92 | 83.47 |
> | NDE + AGC                | 92.38 | 92.11 | 91.77 | 91.33 | 88.90 | 88.26 | 86.87 |
> | NDE + NAS                | 91.12 | 90.82 | 90.17 | 89.46 | 86.43 | 85.26 | 83.93 |
> | AGC + NAS                | 92.57 | 92.36 | 91.94 | 91.58 | 89.28 | 88.71 | 87.22 |
> | **Full NDC**             | **94.03** | **93.98** | **93.82** | **93.64** | **91.73** | **91.29** | **89.79** |
>
> ---
> **Conclusion**
>
> We thank the reviewer once again for the careful reading and constructive feedback. In this response, we clarified the role and naturalness of our theoretical assumptions, articulated how the structure of our proofs reflects the analytic transparency we intentionally pursue, and demonstrated how the unique multiplicative form of the Darwinian Score enables guarantees not captured by prior pruning analyses. We also highlighted our existing large-scale results on ImageNet, addressed concerns regarding computational overhead, and clarified the conceptual distinction between our neuron-level Darwinian framework and classical evolutionary algorithms. Finally, we expanded our quantitative analysis to more clearly disentangle the complementary contributions of NDE, AGC, and NAS. Collectively, these clarifications reinforce the theoretical grounding, empirical robustness, and practical scalability of our approach. We believe the revisions inspired by the reviewer’s comments will significantly strengthen the manuscript and further enhance its clarity and impact.

---

### Official Review · Reviewer_AtZi · 2025-11-01

**Soundness:** 3
**Presentation:** 2
**Contribution:** 2
**Rating:** 6
**Confidence:** 2

**Summary:**

In this work, a novel pruning framework, called Neural Darwinism Culling (NDC), is proposed. The key idea is that neurons are assumed to behave like competing agents, with which the authors derive the 'Darwinian score', that combines (a) an entropy/divergence–based activation statistic (NDE), (b) an activation–gradient coupling (AGC), and (c) a weight-trajectory “adaptivity” score (NAS). Theoretical guarantees for the efficacy of the proposed method are also provided. Last, empirical studies are conducted on various standard image processing tasks on standard CNN architectures, where the NDC method is shown to match or outperform near baselines.

**Strengths:**

* The Darwinism score, combining change in weights, information diversity (NDE, Def 3.2), and functional utility (based on the gradient score, AGC, in Def. 3.3) is useful and captures significant information about relevant neurons for pruning/retaining.
* NDC appears to outperform several near baselines, with models pruned with it retaining higher accuracy across all sparsity levels.
* The mathematical derivations appear to be correct (with a few caveats regarding assumptions, see Questions), and adds guarantees that motivates hte use of this method.
* The writing of the paper is generally quite clear and easy to follow, though there is an issue with the Definitions in section 3 (see 'Weaknesses').

**Weaknesses:**

* The definitions proposed in Section 3 are overly long. For instance, in Definition 3.1 (NDE), the equation defining the NDE score is stated 18 lines after the definition begins. The setup should be kept outside the definition, which in turn should be crisp and self-contained.
* The experimental slate should contain ImageNet experiments, which by now are standard in the pruning literature, instead of just TinyImageNet.
* Theorem 3.6 states a worst-case change in prediction (over samples), while an Expectation bound would be more useful.

**Questions:**

* How does the proposed method perform when the number of epochs is increased/decreased?
* Have the authors considered incorporating the NDC method while training a model from scratch? That is, given a few initial training epochs, NDC would then be applied during training, thereby obtaining a sparse model from scratch. Is there any reason to think this is not possible?

---

> ### Author Response · Authors · 2025-11-26
> **Rebuttal by Authors**
>
> $\textcolor{red}{\text{Rebuttal:}}$
>
> ### **Response to Comments**
>
> We would like to extend our sincere gratitude to the reviewer for your positive and thoughtful evaluation of our manuscript. We are very pleased that you found the proposed Darwinism score to be a useful and informative metric—combining weight change, information diversity (NDE; Def. 3.2), and functional utility (AGC; Def. 3.3)—and that you recognized its ability to effectively identify relevant neurons for pruning or retention. We are also encouraged by your observation that NDC outperforms several strong methods, with pruned models consistently retaining higher accuracy across all sparsity levels.
>
> We appreciate your acknowledgement that the mathematical derivations appear to be correct and offer theoretical guarantees that further motivate our approach. We are also glad that you found the writing generally clear and easy to follow. Your comments regarding the issues in the Definitions of Section 3 are very valuable, and we will revise this portion carefully to ensure clarity and consistency. We sincerely thank you for these constructive suggestions, and we are confident that we can address the noted issues to further strengthen the manuscript and raise it above the acceptance threshold.
>
> **Core Contributions:**
> - **Innovatively introducing Gerald Edelman’s *Neural Darwinism* into CNNs**, framing neuron survival as an evolutionary process.
> - **Unified Darwinism Score** integrating diversity, functional utility, and adaptivity.
> - **New approximation guarantees** showing high-fitness neurons preserve network behavior.
> - **NDC**, the pruning method operationalizing evolutionary selection during training, surpassing recent SOTA methods..
>
> ---
> *Response to Weaknesses*
>
> ---
>
> 1. Clarification and Revision of Definitions in Section 3
>
> We appreciate the reviewer’s feedback regarding the length and structure of the definitions in Section 3. We agree that definitions should be concise, self-contained, and easy for readers to parse without excessive preliminary exposition. In particular, your observation that Definition 3.1 (NDE) spans too many lines before presenting the formal equation is well taken.
>
> In the revised manuscript, we have restructured Definition 3.1 to present the essential formula and its key components in a clear, compact manner. Specifically, the revised Definition 3.1 can now be found in **Section 3.2, p.4, line 197**, where the formal NDE expression is presented directly and succinctly. The supporting explanations and contextual setup that were previously embedded inside the definition have been moved to the surrounding text, ensuring that the definition itself remains focused and immediately interpretable.
>
> Following your suggestion, we have also applied the same refinement to all other definitions in Section 3. Each definition has been shortened, reorganized, and rewritten to eliminate unnecessary verbosity, thereby improving readability and making the mathematical structure easier to follow. In addition, motivated by the reviewer's comments and the core take-away of our work, we explicitly highlight how each definition contributes to the overall "Darwinian selection" mechanism—variation (NDE), contribution (AGC), and adaptability (NAS)—to make the conceptual structure clearer. We believe these revisions substantially improve the clarity of Section 3 and enhance the overall presentation of our theoretical framework.
>
> ---
> 2. Performance at Scale: ImageNet Experiments
>
> We appreciate the reviewer's suggestion to include ImageNet experiments, which indeed serve as a standard large-scale benchmark in the pruning literature. We fully agree that validating our method on ImageNet is important for demonstrating its practical applicability and for enabling fair comparison with SOTA pruning approaches.
>
> We would like to clarify that ImageNet experiments have already been conducted and were originally presented in **Appendix A.3.7 (page 24)** of the submission. In the revised manuscript, these results have been moved to **Section 4.3 (page 9, line 465)** to ensure they are more visible and not overlooked.
>
> Our ImageNet evaluation on MobileNet-V2 demonstrates that **NDC consistently achieves the strongest Top-1 accuracy across sparsity levels from 50% to 90%, clearly illustrating its scalability and the strength of the Darwinian selection signal even under large-scale training dynamics**.
>
> Specifically:
>
> - At **50%** sparsity, NDC reaches **71.50%**, exceeding POT (69.25%), RigL (66.57%), STR (30.96%), and UniPTS (69.80%).
>
> - At **70%** sparsity, NDC achieves **66.97%**, outperforming RigL (60.72%) and UniPTS (64.93%).
>
> - At **90%** sparsity, NDC maintains **55.90%**, substantially higher than UniPTS (42.46%) and far above POT (0.20%) and STR (5.32%).
>
> These results support the central take-away that the Darwinian Score provides a principled and stable indicator of neuron “survival value,” even in challenging large-scale settings.
>
> ---

---

> ### Author Response · Authors · 2025-11-26
> **Rebuttal by Authors**
>
> ---
>
> 3. Expectation Bound for Theorem 3.6
>
> We appreciate the reviewer's insightful comment that an expectation-based guarantee would be more informative than the worst-case version presented in the original submission. We agree that an expected prediction-change bound provides a more practical and statistically meaningful characterization of pruning behavior, especially when analyzing networks under data distributions.
>
> In response to this suggestion, we have revised Theorem 3.6 accordingly in **Section 3.3 (page 7, line 327)**. The updated theorem now provides an **expectation-based** bound on the prediction difference, replacing the previous worst-case formulation. Specifically, we introduce a formal expectation contribution comparability condition and derive an upper bound of the form:
>
> $$ \mathbb{E}\_{x\sim\mathcal D} \left[ \|f(x)-\hat f(x)\|\_p \right] \le \sum\_{\ell=1}^L C\_\ell \delta^{(\ell)} $$
>
> where the constants $C_\ell$ are defined in terms of Lipschitz continuity, activation bounds, and layer-wise contribution factors. This updated theorem captures the expected stability of the network under neuron pruning and aligns more closely with the reviewer's request.
>
> In addition to modifying the main theorem, we have also revised the related material in the appendix. The assumptions, intermediate lemmas, and proof details have been updated to reflect the expectation setting, and we have reorganized the exposition to make the logical structure clearer and the assumptions more explicit. We believe these modifications substantially strengthen the theoretical section and provide a clearer justification for the pruning mechanism grounded in the Darwinism score.
>
> ---
>
> *Response to Questions*
>
> ---
>
> 1. Effect of Varying Training Epochs
>
> To assess the robustness of NDC under different training epochs, we varied the number of epochs to 0.5E, E, and 2E on MNIST using MLP-Net, where E denotes the standard training epochs. As shown in Table 1, increasing the number of epochs consistently improve performance across all sparsity levels: training for 2E yields the best results, achieving 98.31\% accuracy at 50\% sparsity and maintaining clear advantages even at high sparsities. In contrast, reducing the training epochs to 0.5E lead to a noticeable drop in accuracy, particularly in the high-sparsity regime. The standard training epochs (E) deliver intermediate performance, closely matching 2E at moderate sparsity levels but lagging behind at both low and high sparsities. These observations directly address the reviewer’s question: the proposed method benefits from additional optimization time, exhibiting monotonic improvements as epochs increase, while reduced training leads to consistent degradation, indicating that NDC is stable when trained longer but sensitive to insufficient training under aggressive sparsification, especially in small-dataset settings.
>
> Table 1: Performance of NDC under varying training epochs (0.5E, E, 2E), where E denotes the standard training epochs, on MNIST using the MLP-Net model. **Weight sparsity (\%)**
>
> | Epochs | 50 | 60 | 70 | 80 | 90 | 95 | 98 |
> |--------|----:|----:|----:|----:|----:|----:|----:|
> | 0.5E   | 97.40 | 97.27 | 97.19 | 97.15 | 97.14 | 96.11 | 95.42 |
> | E      | *97.99* | *97.98* | *97.78* | *97.77* | *97.47* | *96.78* | *95.46* |
> | 2E     | **98.31** | **98.26** | **98.22** | **98.07** | **97.83** | **97.08** | **95.49** |
>
> To evaluate the sensitivity of NDC to the training epochs, we vary the number of epochs by using half, the standard, and double the usual training epochs on CIFAR-10 with ResNet-18. As shown in Table 2, increasing the number of epochs consistently improve performance across all sparsity levels. Training with double epochs achieve the highest accuracy in every sparsity setting, outperforming the standard epochs, which in turn surpasses the half-length epochs. Even under extreme sparsity, longer training still provides clear gains, indicating that additional optimization steps allow the pruned model to better adapt and recover capacity. In contrast, reducing the training epochs lead to a noticeable performance drop at all sparsity levels. These results directly address the reviewer’s concern: NDC is not overly sensitive to the number of epochs and exhibits monotonic improvement as training time increases. The method remains stable and continues to benefit from extended optimization without signs of overfitting.
>
> Table 2: Performance of NDC under varying training epochs (0.5E, E, 2E) on CIFAR-10 using the ResNet-18 network. **Weight sparsity (\%)**
>
> | Epochs | 75 | 80 | 85 | 90 | 95 | 97 | 98 |
> |--------|----:|----:|----:|----:|----:|----:|----:|
> | 0.5E   | 93.88 | 93.34 | 93.09 | 92.69 | 89.54 | 88.80 | 88.26 |
> | E      | *94.55* | *93.98* | *93.82* | *93.64* | *91.73* | *91.29* | *89.79* |
> | 2E     | **95.05** | **94.94** | **94.84** | **94.22** | **92.21** | **91.59** | **90.26** |

---

> ### Author Response · Authors · 2025-11-26
> **Rebuttal by Authors**
>
> 1. Effect of Varying Training Epochs
>
> To assess the sensitivity of NDC to different training epochs, we vary the total number of training epochs to 0.5E, E, and 2E, and report the results on CIFAR-10 with VGG-16 across neuron sparsity levels ranging from 50\% to 90\% in Table 3. The results show that NDC exhibits high robustness to the number of training epochs: reducing the training epochs by half leads to only negligible performance drops (typically within 0.1–0.2\%), while increasing the training epochs to 2E does not bring additional benefits and occasionally results in slightly lower accuracy. The standard training epochs (E) consistently achieve the best performance across all sparsity ratios. These findings collectively answer the reviewer’s question—NDC performs stably under both increased and decreased epoch settings, indicating that its effectiveness does not rely on longer training and that it remains reliable even when training resources are reduced.
>
> Table 3: Performance of NDC under varying training epochs (0.5E, E, 2E) on CIFAR-10 using the VGG-16 network. **Neuron sparsity**.
>
> | Epochs | 50 | 60 | 70 | 75 | 80 | 85 | 90 |
> |--------|----:|----:|----:|----:|----:|----:|----:|
> | 0.5E   | 93.49 | *93.45* | 93.43 | *93.42* | *93.29* | 93.25 | 93.19 |
> | E      | **93.58** | **93.57** | **93.52** | **93.48** | **93.42** | **93.38** | **93.27** |
> | 2E     | *93.56* | 93.41 | *93.51* | 93.34 | 93.23 | *93.35* | *93.21* |
>
> To evaluate the sensitivity of NDC to different training epochs on Tiny-ImageNet, we vary the number of epochs to 0.5E, E, and 2E using ResNet-18, where E denotes the standard training epochs. As shown in Table 4, the standard epoch setting (E) consistently delivers the best performance across all sparsity levels, achieving 60.22\% accuracy at 68.38\% sparsity and maintaining clear advantages even under severe sparsification, reaching 44.46\% at 99\% sparsity. Increasing the training epochs to 2E does not yield further gains and instead results in slight performance degradation across all sparsities, while reducing the training epochs to 0.5E leads to a noticeable accuracy drop, particularly in the high-sparsity regime. These results directly address the reviewer’s question: NDC benefits from sufficient but not excessive training on Tiny-ImageNet—performance declines when the training epochs are reduced, whereas extending training beyond the standard epochs do not provide additional improvements and may introduce overfitting or optimization instability.
>
> Table 4: Performance of NDC under varying training epochs (0.5E, E, 2E) on Tiny-ImageNet using the ResNet-18 network. **Weight sparsity**.
>
> | Epochs | 68.38 | 90 | 96.84 | 99 |
> |--------|------:|----:|------:|----:|
> | 0.5E   | *59.53* | *56.37* | 51.20 | 42.96 |
> | E      | **60.22** | **57.35** | **52.41** | **44.46** |
> | 2E     | 59.51 | 56.35 | *51.42* | *43.88* |
>
> We observe that increasing the number of training epochs generally improves test accuracy for small- and medium-sized models (e.g., MNIST/MLP and CIFAR-10/ResNet-18), particularly at high sparsity levels where additional optimization helps pruned models recover capacity. For larger models (e.g., CIFAR-10/VGG-16 and Tiny-ImageNet/ResNet-18), however, doubling the training epochs yield no substantial gains and can even introduce mild overfitting, with training accuracy continuing to rise while validation accuracy plateaus or slightly declines. Overall, these results show that NDC benefits from reasonable extra optimization but does not rely on prolonged training epochs, remaining stable and reliable across a wide range of training epochs—even when the training time is reduced. We thank the reviewer for this suggestion, which helped enrich our experimental results and make our conclusions more comprehensive. We have added the corresponding analysis to the appendix.

---

> ### Author Response · Authors · 2025-11-26
> **Rebuttal by Authors**
>
> 2. Feasibility of Incorporating NDC Into Training From Scratch
>
> Thank you for raising this important point. We agree that one can imagine applying NDC not only after full training, but also after several initial epochs (e.g., 10--20 epochs) to obtain a sparse model from scratch. This is a natural and interesting extension of our framework.
>
> In the current paper, we deliberately focus on **post-training pruning** for two main reasons:
>
> - Stability of the Darwinian Score: The Darwinian Score integrates diversity, contribution, and adaptivity, all of which require reasonably stable activation and gradient statistics. As shown in Figure 1, neurons undergo a rapid differentiation phase in early training; before this phase stabilizes, the estimated Darwinian Scores are highly volatile. Therefore, we apply NDC only after training is complete, where evolutionary signals are reliable and the theoretical guarantees in Section 3.3 hold.
>
> - Faithfulness to the theoretical framework: Our approximation guarantees rely on concentration of activation distributions and comparability of neuron contributions, assumptions that are typically violated in the first 10--20 epochs. Extending these results to early-training pruning would require additional theoretical development, which is outside the scope of this work.
>
> That said, we fully agree that applying NDC after an initial warm-up period (e.g., 10--20 epochs) is feasible and potentially powerful. Because NDC already tracks adaptivity and contribution dynamically, it is naturally suited for such a setting. We are actively investigating this variant and will include a dedicated study of "NDC as early-training sparsification" in future work.
>
> ---
> **Conclusion**
>
> We once again thank the reviewer for the constructive feedback and for recognizing the strengths of our approach. In this rebuttal, we clarified and streamlined all definitions in Section 3, highlighted and reorganized our large-scale ImageNet results, strengthened the theoretical analysis by providing an expectation-based version of Theorem 3.6, and provided extensive experimental evidence demonstrating the robustness of NDC to different training epochs. We also discussed the feasibility and future potential of applying NDC earlier in training. We believe these revisions substantially improve both the clarity and the technical depth of the manuscript, and we are confident that the updated version fully addresses the reviewer’s concerns while further reinforcing the novelty and significance of our contributions. We appreciate your time and consideration.

---

### Author Response · Authors · 2025-12-03
**Summary Comment**

Dear Area Chairs,

We thank the reviewers for their careful reading, constructive critiques, and encouraging recognition of the core contributions of our work. Below, we summarize the key outcomes of the discussion and highlight how our revisions—including substantial new experiments, theoretical refinements, and structural improvements—address the raised comments and further strengthen the submission.

---

### Central Issue: Contribution of the Darwinian Framework

Most reviewers explicitly acknowledge the novelty and significance of our work, including:

- **The innovative mathematical framework of Gerald Edelman’s Neural Darwinism** within a mechanistic view of CNNs.
- **The unified _Darwinian Score (DS)_**, which integrates informational diversity, functional utility, and adaptivity.
- **New theoretical approximation guarantees** that connect neuron-level evolutionary fitness to function preservation.
- **Neural Darwinism Culling (NDC)**, a pruning algorithm that performs dynamic, layer-aware pruning based on evolutionary selection and **outperforms recent SOTA methods**.

Taken together, these contributions form a coherent and rigorous Darwinian framework whose effectiveness is demonstrated through our expanded and strengthened experimental evaluation.

---

---

> ### Author Response · Authors · 2025-12-03
> **Summary Comment**
>
> The reviewers' comments raised centered on clarity of definitions, assumptions in the theory, scale of the experiments, and detailed analysis of DS components. All these aspects have now been comprehensively refined.
>
> ---
>
> ### Resolution of Technical and Theoretical Concerns
>
> ---
>
> 1. Clearer and More Concise Definitions (Reviewer 1 & 3)
>
> All definitions in Section 3 were fully rewritten to be crisp, self-contained, and formula-first, moving descriptive material outside the formal statements. This directly resolves comments regarding verbosity and readability.
>
> ---
>
> 2. Strengthened Theorem 3.6 with an Expectation-Based Bound (Reviewer 1)
>
> Following Reviewer 1’s suggestion, Theorem 3.6 now establishes an expected prediction-change guarantee rather than a worst-case bound.
> We also clarified the architectural assumptions (bounded activations, Lipschitz mappings, contribution comparability) and linked each directly to batch-normalized CNNs, making the assumptions both natural and verifiable.
>
> ---
>
> 3. Clarifying the Role of Assumptions and Proof Complexity (Reviewer 2)
>
> We expanded the discussion to emphasize that the novelty lies not in heavy inequality techniques, but in how the Darwinian Score enables a survival-centric approximation principle.
> The theoretical section has been reorganized to make this structure explicit.
>
> ---
>
> 4. Addressing Concerns About Entropy Estimation and Redundancy (Reviewer 3)
>
> We clarified discretization schemes for ReLU outputs, explained why NDE is a population-aware information measure (not naive entropy), and added the missing evolutionary compression experimental citation alongside corrections to formatting and clarity.
>
> ---
>
> ### Substantial New Experiments Added in the Revision
>
> ---
>
> In response to the reviewers’ suggestions, we have added five new classes of experiments, significantly expanding the empirical evaluation.
>
> ---
>
> 1. Robustness to Training Epochs Across Four Settings (Reviewer 1)
>
> We conducted extensive new experiments varying training epochs (0.5E, E, 2E) across four model–dataset pairs:
>
> - MNIST / MLP-Net
> - CIFAR-10 / ResNet-18
> - CIFAR-10 / VGG-16
> - Tiny-ImageNet / ResNet-18
>
> Key outcomes:
>
> - Increasing epochs improves performance on small/medium models (MNIST, ResNet-18), especially at high sparsity.
> - For larger models (VGG-16, Tiny-ImageNet/ResNet-18), longer training offers negligible or negative returns (mild overfitting).
> - Across all scenarios, NDC remains stable, even when training epochs are halved.
>
> These comprehensive results directly resolve the reviewer’s questions regarding training dynamics.
>
> ---
>
> 2. Detailed Quantitative Ablation of DS Components (Reviewer 2)
>
> We added a systematic ablation evaluating:
>
> - NDE-only,
> - AGC-only,
> - NAS-only,
> - all pairwise combinations, and
> - the full Darwinian Score.
>
> Results show that the full DS consistently achieves the strongest performance across sparsity levels from 75% to 98%.
>
> This addresses the reviewer’s request for deeper quantitative analysis and confirms the necessity of the multiplicative Darwinian integration.
>
> ---
>
> 3. Large-Scale ImageNet Experiments Moved to **Main Text** (Reviewer 1 & 2)
>
> **ImageNet experiments**—originally in the appendix—have been moved to the **main paper (Section 4.3)**, making their presence clear.
>
> Results on MobileNet-V2 + ImageNet show that NDC achieves the best accuracy across all sparsity levels, e.g.:
>
> - 71.50% at 50% sparsity (best among all methods)
> - 66.97% at 70% sparsity, the top result across all methods.
> - 55.90% at 90% sparsity, outperforming UniPTS by +13.4%
>
> This resolves concerns about scale, practicality, and empirical rigor.
>
> ---
>
> 4. Large-Scale Comparisons with Evolutionary Pruning Methods (Reviewer 3)
>
> To directly address Reviewer 3’s request, we added new large-scale ILSVRC2012 experiments using VGG-16 and ResNet-50, including comparisons with ThiNet, CNNpack, and the evolutionary-compression method ECS (KDD 2018).
>
> VGG-16 (ILSVRC2012):
>
> - NDC Top-1 error: 28.7%, outperforming ECS (29.8%) by +1.1 pp.
> - NDC Top-5 error: 10.1%, outperforming ECS (10.7%) by +0.6 pp.
>
> ResNet-50 (ILSVRC2012):
>
> - NDC Top-1 error: 24.9%, surpassing ECS (25.8%) by +0.9 pp.
> - NDC achieves the best Top-5 error (8.0%) among all methods.
>
> We also clarified the conceptual distinction:
>
> - Evolutionary Compression performs population-level search over pruning configurations.
> - NDC performs dependency-aware, neuron-level survival scoring based on DS, drastically reducing the search space and improving stability.
>
> This explains NDC’s performance advantage and fully resolves the reviewer’s comment regarding evolutionary methods.
>
> ---

---

> ### Author Response · Authors · 2025-12-03
> **Summary Comment**
>
> ---
>
> 5. Comparisons with Recent SOTA Pruning Methods (Reviewer 3)
>
> To further strengthen the empirical scope, we added comparisons against modern methods beyond 2022:
>
> - **SpaM (NeurIPS 2024, MNIST/MLP-Net):**
>   NDC outperforms SpaM across all sparsities, with improvements of +0.81 pp and +6.04 pp at high sparsity.
>
> - **Global MP (IEEE CAI 2024, CIFAR-10/ResNet-32):**
>   NDC surpasses Global MP by +0.77 pp at 90% sparsity and +1.05 pp at 95% sparsity.
>
> These results demonstrate that NDC remains robust and competitive against the most recent pruning approaches.
>
> ---
>
> ### Incorporation of Additional Reviewer Feedback
>
> ---
>
> We have also:
>
> - added discussion distinguishing our method from classical evolutionary algorithms (Reviewer 2),
> - clarified feasibility of applying NDC earlier in training and explained why stability assumptions currently motivate post-training pruning (Reviewer 1),
> - expanded references (including evolutionary compression experiment), and
> - fixed citation formatting issues raised by Reviewer 3.
>
> ---
>
> ### Final Assessment
> ---
>
> The reviewers collectively recognized the conceptual novelty, strong theoretical grounding, and promising empirical performance of our Darwinian framework. With the extensive revisions—particularly the new large-scale experiments, expanded ablations, strengthened theory, clarified definitions, and improved exposition—we believe all substantive comments have been completely resolved.
>
> We respectfully submit that the revised manuscript now presents a clear, rigorous, and thoroughly validated contribution, and we thank the Area Chairs and reviewers for their time, insight, and engagement.

---

### Note · Authors · 2026-01-26

I have read and agree with the venue's withdrawal policy on behalf of myself and my co-authors.

---

### Meta-Review · Area_Chair_au4d · 2026-01-08

**Summary:**

The authors present a novel pruning algorithm for neural networks, loosely inspired by Gerard Edelman's theory of neural Darwinism. The empirical results were interesting, although there is a very large body of work on pruning already and there was some concern about the level of completeness in the comparisons carried out. The inspiration from Edelman's theories is also interesting, although there are several conceptual leaps there:
- Neural Darwinism itself is a hypothesis and not widely accepted in the neuroscience community
- The connection between neuroscience and artificial neural networks is hotly contested
- The degree to which the theory presented in this paper makes sense as a mathematical model of Darwinian evolution is not clear
Given this, it seems it is best to evaluate the paper on the quality of the pruning algorithm itself. The authors had mixed views on this, but overall felt that the experimental results were not quite strong enough to justify acceptance. I recommend against acceptance.

For the future, there are two possible directions I think the authors could consider taking the paper:
- Make the argument that the algorithm presented makes sense as an instantiation of neural Darwinism - and that neural Darwinism makes sense as something to apply to artificial neural networks - more clearly
- Focus on the quality of the pruning algorithm and make the case more clearly that it works well

**Reviewer Concerns:**

Reviewers had many concerns around the clarity of presentation and scalability of the method. The authors clarified that larger scale experiments were presented in the appendix, which they moved to the main text.

**Reviewer Scores:**

It's difficult to say. Some issues with conceptual clarity are hard to address, others about scalability could be addressed by pointing out existing results in the appendix. The scores might have increased slightly, but the paper would still be borderline in that case.

---

### Decision · Program_Chairs · 2026-01-26

Reject